# Injectable and biodegradable piezoelectric hydrogel for osteoarthritis treatment

Tra Vinikoor [1,2], Godwin K. Dzidotor[2,3], Thinh T. Le[4], Yang Liu [5], Ho-Man Kan[2], Srimanta Barui[2], Meysam T. Chorsi [4], Eli J. Curry [6,16], Emily Reinhardt[7], Hanzhang Wang[8], Parbeen Singh[4], Marc A. Merriman [2,3], Ethan D'Orio [9], Jinyoung Park [1], Shuyang Xiao[10], James H. Chapman [2], Feng Lin[4], Cao-Sang Truong[1], Somasundaram Prasadh[11], Lisa Chuba[12], Shaelyn Killoh[12], Seok-Woo Lee [10,13], Qian Wu[8], Ramaswamy M. Chidambaram[12], Kevin W. H. Lo[2,13,14], Cato T. Laurencin [1,2,3,10,13,15] & Thanh D. Nguyen [1,4,13] ✉

Osteoarthritis affects millions of people worldwide but current treatments using analgesics or anti-inflammatory drugs only alleviate symptoms of this disease. Here, we present an injectable, biodegradable piezoelectric hydrogel, made of short electrospun poly-L-lactic acid nanofibers embedded inside a collagen matrix, which can be injected into the joints and self-produce localized electrical cues under ultrasound activation to drive cartilage healing. In vitro, data shows that the piezoelectric hydrogel with ultrasound can enhance cell migration and induce stem cells to secrete TGF-β1, which promotes chondrogenesis. In vivo, the rabbits with osteochondral critical-size defects receiving the ultrasound-activated piezoelectric hydrogel show increased subchondral bone formation, improved hyaline-cartilage structure, and good mechanical properties, close to healthy native cartilage. This piezoelectric hydrogel is not only useful for cartilage healing but also potentially applicable to other tissue regeneration, offering a significant impact on the field of regenerative tissue engineering.

Osteoarthritis (OA) is a prevalent joint disease that results in the deterioration and damage of cartilage tissue, which can lead to extreme pain and significantly limit the daily functioning of those affected. It is estimated that approximately 654 million people worldwide are impacted by OA every year[1]. Current non-surgical treatments using analgesics and anti-inflammatory drugs only alleviate symptoms and do not completely cure the disease[2,3]. Meanwhile surgical interventions such as subchondral bone microfracture, lavage, debridement, and shaving are only effective for small chondral defects[4,5]. For severe articular lesions, the gold standard treatment is to surgically implant autologous or allogenic cartilage grafts. However, these grafts experience the complications of donor site morbidity, immune-rejection, infection, and especially, limited tissue supply[6,7]. Therefore, regenerative engineering approaches, which utilize

biomaterial scaffolds to construct engineered cartilage grafts, have emerged as an important field[8,9]. Despite many encouraging results, the clinical use of engineered graft cartilages has been restricted due to (1) the toxicity of the additional growth factors to promote cartilage healing, (2) the inefficiency of stem cell differentiation, and (3) the invasive surgical process to implant the grafts[10,11].

Electrical stimulation (ES) has been known to exhibit a significant effect on promoting tissue regrowth, especially for bone and cartilage[12–14]. Within cartilage, there are electrical currents/charges that are naturally generated during joint movements or deformations[15,16]. Thus, cartilage repair may benefit from external ES, which supplements the disrupted electrical microenvironment in damaged cartilages. Indeed, ES has been proven to induce mesenchymal stem cells to differentiate into chondrocyte cells in the absence of essential growth factors in vitro[12,17,18]. Bakers et al. has shown that implanting a battery

and electrodes into cartilage defects in rabbit knees could promote the regeneration of hyaline cartilage[19]. A similar result was also reported when microcurrent stimulation was utilized to heal cartilage on both immature and adult rats[20,21]. Despite the regenerative capability of ES on articular cartilage, the current approaches for ES experience a variety of issues. External application of ES can be significantly weakened by the absorption of surrounding tissues, which can limit its effectiveness[22,23]. Although implanting ES devices can be a strategy, it poses significant risks of infection, and the reliance on toxic batteries in this approach requires invasive removal procedures that may disrupt the healing tissue[24].

Piezoelectric materials possess the ability to self-produce electricity under mechanical stress such as joint load or acoustic pressure of ultrasound (US). Due to this unique property, piezoelectric materials can be appealing candidates for self-powered electrical stimulators that eliminate the need for toxic batteries. Indeed, these smart materials have been significantly utilized in various biomedical applications including implantable sensors, ultrasound transducers, actuators[25–30]. In the field of tissue engineering, piezoelectric materials have demonstrated their potential for bone, cartilage, skeletal muscles, skin, and nerve regeneration[31–35]. For example, after corona poling, a scaffold made of out poly(vinylidene fluoride) (PVDF) favored myoblast cell adhesion and growth due to its negative surface charges[36]. Electrospinning poly(vinylidene fluoride-trifluoroethylene) (PVDF-TrFE) and barium titanate (BaTiO$_3$) created a flexible membrane for wound healing that reduced animal recovery time from 14 to 9 days[37]. However, medical usage of such traditional piezoelectric materials is constrained by non-degradability (e.g., PVDF) or the existence of hazardous substances, such as lead in lead zirconate titanate (PZT)[38,39]. Alternative lead-free piezoelectric materials including BaTiO$_3$, potassium sodium niobate (KNN), and Zinc oxide (ZnO) have been introduced[40,41]. Yet, the safety profile and degradation byproducts of these materials are not well-established or validated for long-term use inside the human body[42].

Biodegradable organic piezoelectric materials, including synthetic polymers (e.g., poly-L-lactic acid (PLLA)) and amino acids (e.g., glycine, diphenylalanine), with excellent safety profiles, have recently emerged as alternatives for conventional piezoelectric materials[28,43–47]. Despite the high piezoelectric performance, glycine crystals suffer from a significant problem of rapid degradation (~few hours) in aqueous environment. Therefore, glycine crystals cannot serve as a long-term implanted scaffold to support cell ingrowth and tissue remodeling. In contrast, piezoelectric PLLA has a long degradation time (~1–2 years) and is already a well-known biocompatible material used for many FDA-approved implants or tissue-scaffolds[44,47,48]. Our previous work has demonstrated that electrical charges, generated by the solid PLLA nanofiber scaffold under joint load, can promote the healing of severe osteochondral cartilage defects in rabbits' knees[44]. However, invasive surgical procedures were necessary to implant such piezoelectric PLLA scaffolds. This implantation intervention could cause complications such as damage to other healthy joint tissues, infection, and inflammation along with a long recovery time. In this regard, hydrogels have received a lot of interest in the tissue engineering field, especially for OA treatment, because they can be non-invasively injected into the body, and easily filled into the irregular shapes of defects. Furthermore, they offer a porous aqueous environment to facilitate cell ingrowth and reform damaged tissues[49–54]. Yet, there have not been any reports on an injectable piezoelectric hydrogel that can be remotely activated to deliver electrical cues and then degrade into completely safe, excretable byproducts after healing the damaged tissues.

Here, we introduce a biodegradable injectable piezoelectric hydrogel, made of the cryo-sectioned piezoelectric short nanofibers of PLLA (so-called NF-sPLLA) and collagen matrix (Fig. 1a), which can be (1) inoculated into the joint cartilage defects (visualized by X-Ray or arthroscopic camera, Supplementary Movie 1, Fig. 1b) to avoid invasive implantation surgery and (2) activated by US to electrically stimulate the healing of severe cartilage defects for the treatment of OA.

This piezoelectric hydrogel can produce localized electrical charges under external US to promote cell migration, attract host cells, and induce endogenous growth factors like TGF-β1. This leads to the chondrification process and eventually the deposition of extracellular matrix (ECM) proteins to heal cartilage tissue. Our in vitro study shows that culturing adipose-derived stem cells (ADSCs) inside piezoelectric hydrogel under US treatment induced 9.4-fold increase of *COL2A1*, 10.6-fold increase of *ACAN*, and 12.1-fold increase of *SOX9* gene expression compared to the control groups without piezoelectric effect or US activation. Notably, in a rabbit critical-size osteochondral defect model, we find significant regeneration of hyaline cartilage along with subchondral bone within the defects with strong mechanical properties for the animals receiving our piezoelectric hydrogel after 1–2 months of US activation.

These results demonstrate that piezoelectric NF-sPLLA hydrogel is an important biomaterial that can be injected into the body to heal severe cartilage defects. The hydrogel along with the US activation approach is potentially applicable to the regeneration of other damaged tissues such as bone, nerve, muscle, skin etc. Furthermore, this hydrogel can be used as a platform to create other medical devices such as injectable piezoelectric actuators, sensors, energy-harvesters, and transducers[45,55], thus offering a significant impact not only on the field of tissue regenerative engineering but also potentially on other fields such as drug delivery, health monitoring, and other disease treatments/preventions.

## Results and discussion
### Characterization of injectable piezoelectric NF-sPLLA hydrogel
Our previous studies reported the use of electrospinning PLLA piezoelectric nanofiber mats for multiple purposes[44,45,47,55]. Here, we also deployed the same electrospinning technique to produce piezoelectric PLLA nanofiber mats (Supplementary Fig. 1). Chloroform was used as the solvent due to its fast evaporation rate, allowing nanofibers to form separately. The separated fibers in the mat allowed us to cut and collect individual NF-sPLLA for creating the piezoelectric hydrogel. The collected piezoelectric PLLA nanofiber mat was annealed to increase the crystallinity (see Methods). The mat was then embedded inside an optimal cutting temperature (OCT) embedding medium and cryo-sectioned into short fibers (~25 μm). The samples were cleaned with distilled water and then lyophilized to remove the water. A nuclear magnetic resonance (NMR) test was deployed to certify the purity of collected short fibers. Supplementary Fig. 2a showed there was no peak for chloroform (7.26 ppm, which was used as the solvent to dissolve PLLA for electrospinning) presented in the $^1$H NMR spectra[56]. Also, the NF-sPLLA's pattern $^1$H NMR is similar to the virgin PLLA pellet in our previous study[55]. Thus, we confirmed that no solvent or other toxic chemical residuals were on the NF-sPLLA.

For each PLLA nanofiber, two important parameters that directly influence the piezoelectric property are the β-form crystal structure and the crystallinity[57]. To evaluate these properties, we performed one-dimensional X-ray diffraction (XRD) and differential scanning calorimetry (DSC) on NF-sPLLA and compared them to the PLLA nanofiber film (named Film PLLA). Fig. 1c shows that the XRD diffraction of NF-sPLLA was similar to the original film, and predominantly presented at (200) and (110) crystal faces, which indicates the existence of β-form structures and piezoelectricity[46]. Meanwhile, DSC data indicates that both NF-sPLLA and film PLLA have ~85% crystallinities (Fig. 1d). From both XRD and DSC results, we confirmed that cryo-sectioning nanofiber films into NF-sPLLA did not change material properties.

We employed poly (D, L-Lactide) (PDLLA) as a control material due to its non-piezoelectric property. Although PDLLA and PLLA share similar chemical constituents, PDLLA's structure contains alternating

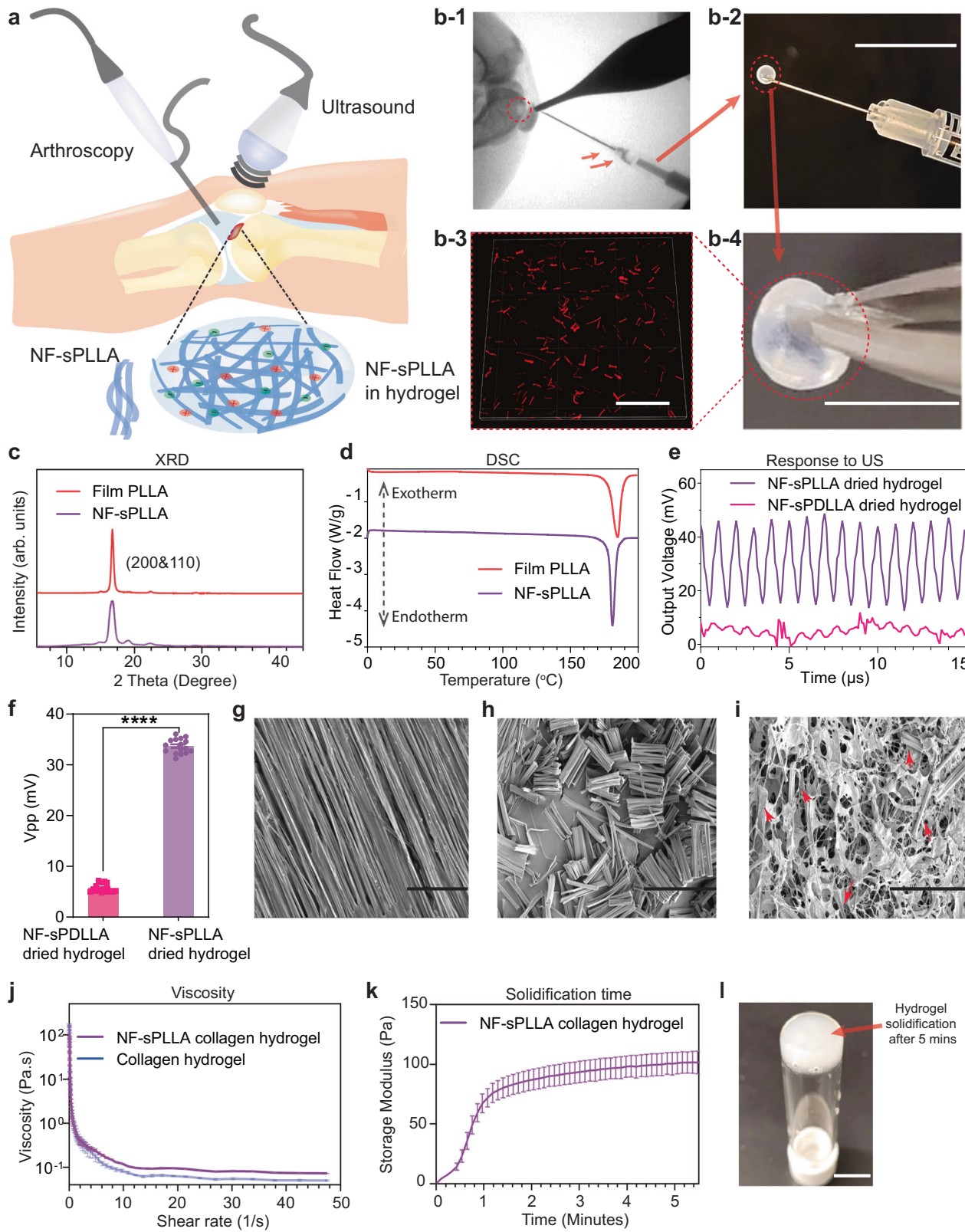

D-lactic and L-lactic acid groups that counterbalance the overall polarization. Consequently, PDLLA does not exhibit piezoelectric behavior, making it an appropriate material for our purpose[55,58]. The PDLLA was electrospun, processed, and cryo-sectioned into short fibers using the same process that was used for NF-sPLLA. Both the piezoelectric NF-sPLLA and non-piezoelectric NF-sPDLLA were mixed with collagen I. In order to verify that NF-sPLLA were uniformly

distributed within the collagen hydrogel, we mixed rhodamine B dye into the PLLA mat, processed and incorporated stained NF-sPLLA in collagen hydrogel, and examined them using a fluorescence microscope. Fig. 1b-1–b-3 confirmed that NF-sPLLA (red) were evenly distributed throughout the hydrogel.

Next, we evaluated the output voltage of the short fibers under US activation. To do that, we vacuum-dried the NF-sPLLA and NF-sPDLLA

**Fig. 1 | Characterization of injectable piezoelectric hydrogel. a** Schematic illustration of the use of piezoelectric hydrogel for OA patients. The piezoelectric hydrogel contains piezoelectric short nanofibers of PLLA (NF-sPLLA) and a hydrogel matrix of collagen, which could be injected into knee joints by arthroscopy or X-ray guidance. The piezoelectric hydrogel is activated by an external US device to generate electrical cues. **b-1** Visualization of a knee defect (red dash circle) on a rabbit cadaver under X-ray and the delivery of our injectable piezoelectric hydrogel. **b-2** Loading piezoelectric hydrogel using insulin G29 needle to test injectability (Scale bar: 1 cm). **b-3** Fluorescent image of rhodamine B stained NF-sPLLA (red) to visualize the distribution of the short nanofibers inside collagen I matrix (Scale bar: 100 μm). **b-4** Macroscopic image of solidified piezoelectric hydrogel (Scale bar: 1 cm). **c** One-dimensional XRD of nanofiber film of PLLA (named as Film PLLA) before sectioning and NF-sPLLA after sectioning (arb. units = arbitrary units). **d** DSC results of PLLA nanofiber film and NF-sPLLA. **e** Output voltage waveform of sensors made of our dried NF-sPLLA hydrogel scaffold (Piezo sample) and NF-sPDLLA

hydrogel (Non-piezo sample) in collagen under US activation ($n = 4$ for independent sensors). **f** Peak-to-Peak output voltage of sensors made of our dried scaffold NF-sPLLA (Piezo sample) and NF-sPDLLA (Non-piezo sample) in collagen under US activation ($n = 4$ for independent sensors, each sensor is measured one time with 4 data points collected, ****$p < 0.000001$, $T$ test, Two-tailed, the data are expressed as Mean ± SEM value). **g** SEM image of film PLLA before sectioning (Scale bar: 40 μm). **h** SEM image of NF-sPLLA after sectioning (Scale bar: 40 μm). **i** SEM image of NF-sPLLA (red arrows) in dried collagen scaffold (Scale bar: 40 μm). **j** Viscosity measurement of NF-sPLLA in collagen hydrogel and collagen hydrogel only ($n = 3$ independent samples, the data are expressed as Mean ± SD value). **k** Time sweep testing to determine the gelation time of NF-sPLLA collagen hydrogel at 37 °C ($n = 3$ independent samples, the data are expressed as Mean ± SEM value). **l** Photograph of upside-down vial of the solidified piezoelectric NF-sPLLA hydrogel that formed a stable structure at 37 °C after 5 mins (the experiment was repeated four times with similar results). Exact $p$ value were provided in the Source Data file.

hydrogel overnight to achieve solid dry scaffolds. Two electrodes were then placed on two sites of the scaffolds to measure the output voltages under US stimulation (see Methods section). Fig. 1e demonstrates the representative output voltage waveforms of the NF-sPLLA dried hydrogel (piezoelectric) and NF-sPDLLA dried hydrogel (non-piezoelectric) sensors. The NF-sPLLA dried hydrogel sensor generates a clear signal with consistent intervals and peak magnitude. Meanwhile, NF-sPDLLA dried hydrogel sensor's waveform has smaller amplitude and irregularity with random peaks under the same applied US intensity. In addition, as shown in Fig. 1f, the output voltage of NF-sPLLA dried hydrogel scaffold is ~33.7 mV peak-to-peak, and is superior to the negative control NF-sPDLLA dried hydrogel scaffolds (~5 mV peak-to-peak). Therefore, we employed PDLLA as a negative control in our consequent in vitro and in vivo studies.

We also examined the morphology of PLLA mat fibers before and after cryo-sectioning using scanning electron microscopy (SEM). The photograph displays the PLLA mat with separate PLLA nanofibers (Fig. 1g) which allowed us to cut and successfully collect the NF-sPLLA (Fig. 1h). Fig. 1i shows the microstructure of NF-sPLLA hydrogel in the dried form with the NF-sPLLA (red arrows) distributed homogenously inside the collagen matrix. Interestingly, we found that incorporated NF-sPLLA inside collagen hydrogel also created more porosity compared to the original collagen gel (Supplementary Fig. 2b, c). This result is consistent with swelling testing that NF-sPLLA-based collagen hydrogel retained much higher amounts of water compared to the collagen hydrogel without the fibers (Supplementary Fig. 2d). This property plays an essential role in tissue engineering as increasing porosity is beneficial to facilitate diffusion of nutrients and oxygen which directly influent cellular ingrowth and tissue remodeling[59]. Based on this finding, we believe that adding NF-sPLLA to the collagen not only provides piezoelectricity to the scaffold, but also creates a favorable 3D-structure for cells/tissues to proliferate.

As shear-thinning is the most crucial property for injectable hydrogel we utilized rheometer to characterize its viscosity. As shown in Fig. 1j, the viscosity of collagen hydrogel with and without NF-sPLLA sharply decreases with the increasing shear rate, suggesting that these hydrogels exhibit shear-thinning behavior[60–62]. The shear-thinning characteristic of the hydrogel remained after adding the NF-sPLLA. To demonstrate the injectability of the NF-sPLLA hydrogel, we loaded NF-sPLLA hydrogel into 1 ml–G29 insulin syringe needle and eventually injected the hydrogel out (Fig. 1b-2 and Supplementary Movie 2). To determine the gelation time of the NF-sPLLA hydrogel, we performed a procedure to select appropriate parameters such as strain and frequency of the shear applied on the hydrogel. We first ran strain sweep testing to determine the linear strain region. A strain sweep from 0.01 to 100% strain was performed at 1 Hz on the solidified hydrogels. Supplementary Fig. 2e indicates that both piezoelectric and pure collagen hydrogels had a linear region up to 10% of strain. Subsequently, we selected 10% strain for the frequency sweeps. Results from

Supplementary Fig. 2f showed that frequency below 10 Hz provided a steady modulus. Therefore, 1 Hz was chosen as the value for the time sweep test. According to the time sweep test, the collagen hydrogel with added NF-sPLLA underwent rapid solidification and attained a structure at around 90 Pa within 3 minutes (Fig. 1k, l).

We also performed the degradation study of the NF-sPLLA hydrogel at 37 °C over a period of 9 weeks, following an accelerated degradation condition (80 °C). As seen in Supplementary Fig. 2g, the volume of the NF-sPLLA hydrogel scaffolds gradually decreased over time. This phenomenon is expected as the hydrogel consists of collagen I and NF-sPLLA, both widely recognized as biodegradable materials[55,63]. Additionally, this result is consistent with our previous research which demonstrated that scaffolds fabricated from collagen I and PLLA fibers are biodegradable[44]. Under the accelerated conditions, the hydrogels quickly degraded, broke down, and lost their original structures.

## Piezoelectric hydrogel for chondrogenesis in vitro study

We hypothesized that piezoelectric hydrogel under external US activation would generate useful electrical charges to promote the chondrogenic differentiation of stem cells. In this regard, both adipose-derived stem cells (ADSC) and bone marrow stem cells (BMSCs) are common mesenchymal stem cells in cartilage regeneration. Both of the cells possess equivalent potential for differentiation into various tissue lineages, including cartilage, bone, and skeletal muscle[64–66]. Despite BMSCs being the local regenerative stem cells, we decided to choose ADSCs because ADSCs offer distinct advantages, including their availability, accessibility, and ease to be expanded for in vitro experiments[67]. More importantly, ADSCs were chosen as the stem cell model to validate our hypothesis that piezoelectric charge can induce mesenchymal stem cells (in general) into a chondrocyte phenotype in vitro (in comparison with the other control groups of using non-piezoelectric stimulation). Therefore, as long as we use the same mesenchymal stem cell and the same conditions for all in vitro groups, our experimental outcomes still serve our purpose to indicate the effect of piezoelectric stimulation on chondrogenesis.

While high frequency 1–3 MHz US has been commonly employed in medical imaging and diagnostic applications, its limited penetration depth in the body makes it unsuitable for accessing knee cartilage or bone[68]. Additionally, prolonged application at these frequencies may result in heat accumulation, potentially causing undesired damage to local tissue[69–71]. To penetrate through knee joint and activate the piezoelectric properties of NF-sPLLA hydrogel, a low frequency (e.g., 40 kHz) is more suitable. This is because a lower tissue absorption rate is observed at a lower frequency[72]. Regardless of the frequency employed, it is crucial to ensure that the US intensity remains below 0.5 Watt/cm$^2$, as the low-intensity US is considered safe for human use[73–76]. As such, we deployed low-intensity 40 kHz ultrasound for our study.

The piezoelectric hydrogel was composed of 5 mg/ml NF-sPLLA, rat tail collagen I solution (3 mg/ml in acetic acid), and 10× PBS, naturalized by 1 N sodium hydroxide. For simplicity, we designated NF-sPLLA-based hydrogel, NF-sPDLLA-hydrogel and collagen as Piezo, Non-Piezo, and control respectively. The results in Supplementary Fig. 3a, b indicated that these hydrogels are highly safe with a low hemolysis rate (<5%) to satisfy the requirements of the International Standards Organization. Supplementary Fig. 3c revealed that both Piezo and Non-Piezo groups are biocompatible with ADSCs in both short-term (1–3 days) and long-term (14 days) assessments. Indeed, the viability of ADSCs in the Piezo + US group was significantly increased on day 9 and day 14 compared to other control/sham groups. This result is consistent with literature indicating that piezoelectric charges/ES had a positive influence on cell growth[77–79].

We next investigated the chondrogenesis of ADSCs seeded within the piezoelectric hydrogel under US activation. It is noteworthy that our intention in using US is to activate electrical charges from piezoelectric hydrogel rather than the direct influence of US on cartilage itself. ADSCs were cultured in the normal growth media and treated with US for 20 mins per day over 14 days. The time period of 20 mins of US treatment was kept constant throughout in vitro and in vivo experiments. We evaluated the ability of piezoelectric hydrogel with US on enhancing chondrogenesis, at both the gene and protein levels. On the gene level, we selected COL2A1, ACAN, SOX9 markers to assess chondrogenesis in vitro, because they are known to be crucial for cartilage tissue. As clearly seen in Fig. 2a–c, after 14 days of the cell stimulation, the 3 genes of COL2A1, ACAN, and SOX9 expression in the Piezo + US group were 9–12-fold higher than the control (collagen only) group. On the other hand, the other groups, including Non-Piezo with and without US, as well as Piezo without US, exhibited only partial upregulation of individual genes (SOX9, ACAN, or COL2A1), but not all three genes, compared to the control group. These results are consistent with previous research, where the presence of fibers alone or solely introducing mechanical or US activation did not enhance chondrogenesis[44,80]. In chondrogenesis studies, when evaluating the ability of a biomaterial to promote cartilage formation, it is important to observe an increase in all these genes. Indeed, our data indicates that the Piezo + US group significantly upregulated all COL2A1, ACAN, and SOX9 genes. Therefore, we believe that the Piezo + US combination provides the best conditions for promoting chondrogenesis. We observed a similar chondrogenic outcome when we repeated the experiment in the chondrogenic media (Supplementary Fig. 4a–c) which is composed of DMEM, sodium pyruvate, dexamethasone, ascorbic acid 2-phosphate, insulin-transferrin-selenite premix, and TGF-β3[44].

At the protein level, GAG and Collagen II are two major proteins in the cartilage matrix, which play essential functions in this tissue. While GAG supports the resistance against compressive loading and cell signaling, collagen II fibers endow elasticity and strength for the cartilages to cushion joint bones under loading[81,82]. As shown in Fig. 2d, e, and Supplementary Fig. 4d the cells seeded on piezoelectric hydrogel with US activation synthesized the most amount of GAG in both normal growth and chondrogenic mediums. Fig. 2f depicts the immunofluorescence staining of collagen type II in various groups. It is evident that the experimental group of Piezo + US exhibited a greater amount of type II collagen synthesis compared to the other control/sham groups either without the piezoelectric effect or without US activation. The results indicated that the local piezoelectric charges, generated from NF-sPLLA hydrogel under US activation, play an important role in inducing stem cell to chondrogenic differentiation. These findings are also consistent with other published works using electrical cues for hyaline cartilage regeneration[17,44,83,84].

We further explored the influence of different amounts of NF-sPLLA in our hydrogels for chondrogenesis. In this experiment, we studied 3 concentrations of NF-sPLLA, 1, 5, and 10 mg/ml in collagen, and the group of cells seeded in the collagen without NF-sPLLA served as the control group. This experiment was performed on ADSCs cultured in the normal growth medium without FBS. The results, as indicated in Fig. 2g–i, suggest that different concentrations of NF-sPLLA in the hydrogel regulate cell behaviors differently. At a higher amount of NF-sPLLA, there were more ACAN and SOX9 generated (increasing from 1.5 to 10-fold, from 1 mg/ml to 10 mg/ml). However, for COL2A1 gene, 5 mg/ml of the NF-sPLLA generated more COL2A1 genes compared to other groups.

In the process of regenerating articular cartilage, it is not only necessary for stem cells to differentiate into chondrocytes but also for them to stably maintain the hyaline cartilage stage, which is distinct from the growth plate zone. Recent studies have revealed that COL2A1 also functions as an extracellular signaling molecule capable of significantly suppressing chondrocyte hypertrophy which promotes integrin β1−SMAD1 interaction[85–87] thereby avoiding cartilage calcification. Also, COL2A1 is considered as an important extracellular signaling molecule that can regulate chondrocyte proliferation and metabolism, similar to soluble signals[88,89]. Furthermore, the collagen network plays a vital role in retaining proteoglycans within the cartilage matrix and is the most essential protein in the hyaline cartilage matrix[90].

In addition to the gene expression, we assessed the piezoelectric performance of the hydrogel at different NF-sPLLA concentrations under US activation (40 kHz and 1 MHz). Supplementary Fig. 5 also shows that 5 mg/ml of NF-sPLLA in the hydrogel generated a significantly higher output voltage compared to the 1 mg/ml and 10 mg/ml concentrations under both 40 kHz and 1 MHz US activation. This result indicates that low amounts of NF-sPLLA in the hydrogel produce minimal piezoelectric charges, therefore showing little to no effect on chondrogenesis. On the other hand, an excessive amount of NF-sPLLA within the same volume of hydrogel leads to a high density of fibers. This high fiber density increases the hydrogel mass, reducing the vibration under US activation, and likely causes the charges generated by the fibers to cancel each other out, leading to a reduction in the overall piezoelectric output voltage. Collectively, 5 mg/ml of NF-sPLLA hydrogel, which provided the best overall piezoelectric performance and the most COL2A1 gene expression, was selected for all subsequent experiments and in vivo studies.

## Activated piezoelectricity induces chondrogenesis by stimulating stem cells to secrete endogenous TGF-β1

Studies have shown that electrical stimulation promotes chondrogenesis by inducing cell migration and the release of TGF-β1, which promotes chondrogenesis[17,37,91,92]. Thus, we hypothesized that the electric cues generated by the hydrogel under US activation could have the same working mechanism (Fig. 3a). To validate our hypothesis, we studied cell migration, TGF-β1 gene expressions, and influence of TGF-β1 blocker on chondrocyte gene markers. We again used ADSCs as the cell model for this mechanistic study.

For cell migration, ADSCs were seeded in glass bottom dishes. Once cells were confluent, we created in vitro wounds by scratching the cell layers with a consistent pipette, and filled the wound bed with Piezo, Non-Piezo, and control samples. Fig. 3b displays the fluorescent images of cells distribution in different groups at 0 h (when wounds were created) and 24 hours afterward. Clearly, the Piezo + US group's defect was mostly covered by migrated ADSCs cells, meanwhile the other groups still exhibited recognizable wound beds. This indicates the piezoelectric charges can indeed accelerate cell migration.

TGF-β1 plays a critical role in signaling stem cell differentiation into chondrocytes from early to mature cartilage stages[93–97]. A report suggested that human chondrocytes cultured in the presence of TGF-β1 increase the production of ECM proteins (including hyaluronan and aggrecan) and eliminate MMP1 and MMP3 protein synthesis[98]. As shown in a previous study, mesenchymal stem cells self-secrete TGF-β1 under ES and differentiate into chondrocytes[17]. Moreover, TGF-β1 can

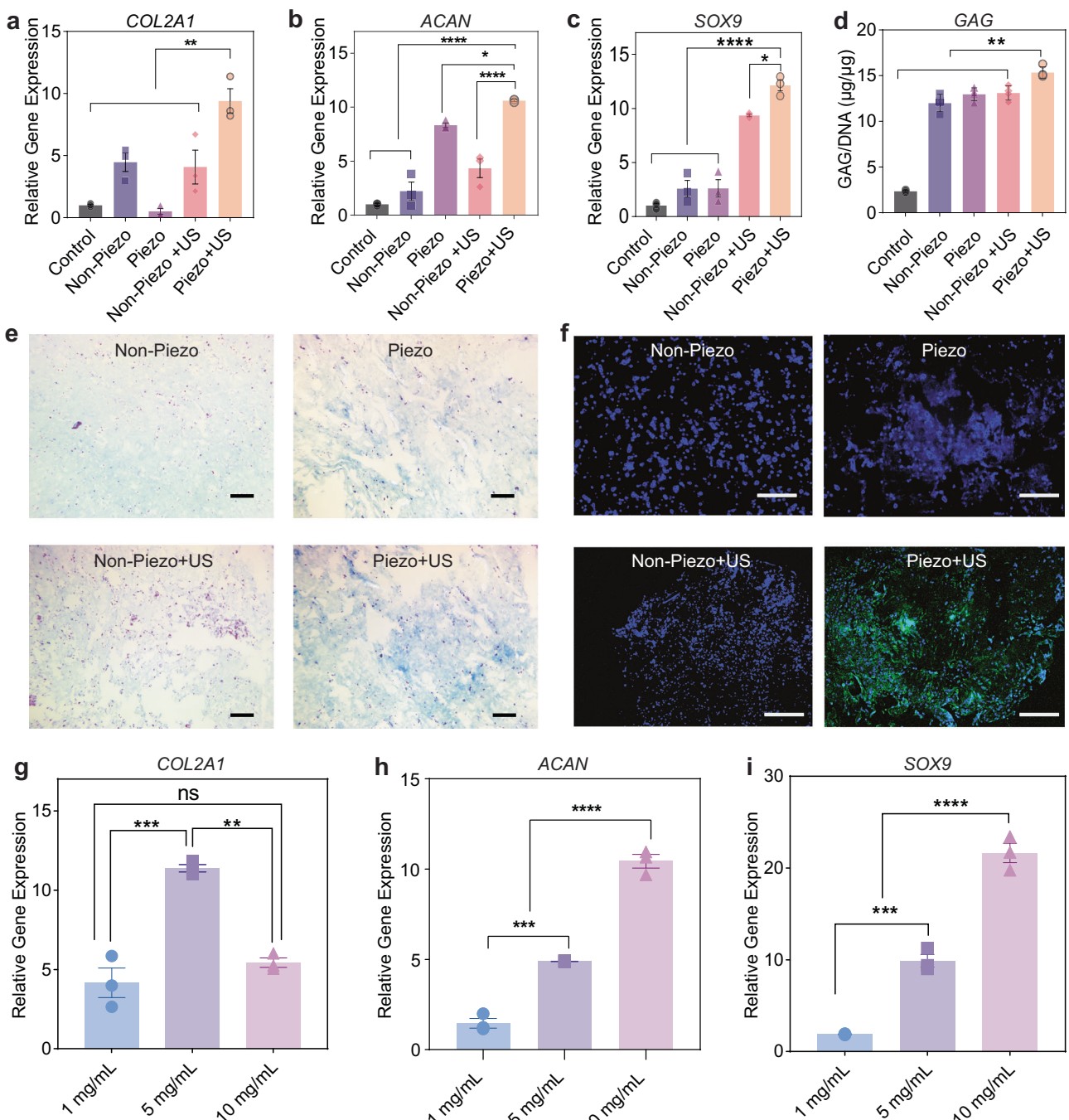

**Fig. 2 | Piezoelectric hydrogel for chondrogenesis in vitro study. a–c** Relative gene expression of the chondrogenic gene markers *COL2A1*, *ACAN*, and *SOX9* (*n* = 3 independent samples, the data are expressed as Mean ± SEM value. **p* < 0.05, ***p* < 0.01 and *****p* < 0.0001, one-way ANOVA, Dunnett's multiple comparisons test). **d** GAG/DNA (µg/ µg) ratio carried out by dimethyl methylene blue (DMMB) kit and dsDNA qualification kit, (*n* = 4 independent samples, the data are expressed as Mean ± SEM value ***p* < 0.01, one-way ANOVA, Dunnett's multiple comparisons test). **e** Alcian blue staining of GAGs (blue) and nuclei (pink) after 2 weeks of culturing cells on different hydrogels (Scale bars: 200 µm). **f** Type II collagen visualization with Immunofluorescence (green) and nuclei (blue) on different hydrogel (Scale bars: 200 µm). **g–i** Relative gene expression of the chondrogenic gene markers *COL2A1, ACAN*, and *SOX9* respectively of ADSCs in the piezoelectric hydrogels with different concentrations of NF-sPLLA and activated by US (*n* = 3 independent samples, the data are expressed as mean ± SEM value, ***p* < 0.01, ****p* < 0.001 and *****p* < 0.0001, one-way ANOVA, Tukey's multiple comparison tests). Exact *p* value were provided in the Source Data file.

upregulate *ACAN* and *COL2A1* gene manufacturing on chondrocyte cells by triggering the Akt/mTOR (the mammalian target of rapamycin) signaling pathway[99]. Hence, we implemented *TGF-β1* gene expression study. Fig. 3c illustrates that cells cultured on the activated Piezo group exhibited the highest amount of *TGF-β1* mRNA after 2 days of piezoelectric stimulation. Furthermore, the piezoelectric hydrogel with US activation can highly suppress the inflammation cytokine of TNF-alpha

compared to the other control/sham groups without the nanofibers, the US activation, or the piezoelectric effect (Supplementary Fig. 6). This could be attributed to the secreted TGF-β1 which is known to inhibit TNF-alpha production.

To further validate the role of endogenous TGF-β1 on chondrogenesis, we used SB431542, a TGF-β inhibitor, to condition the cell medium[100]. Once the TGF-β inhibitor was added into the culture

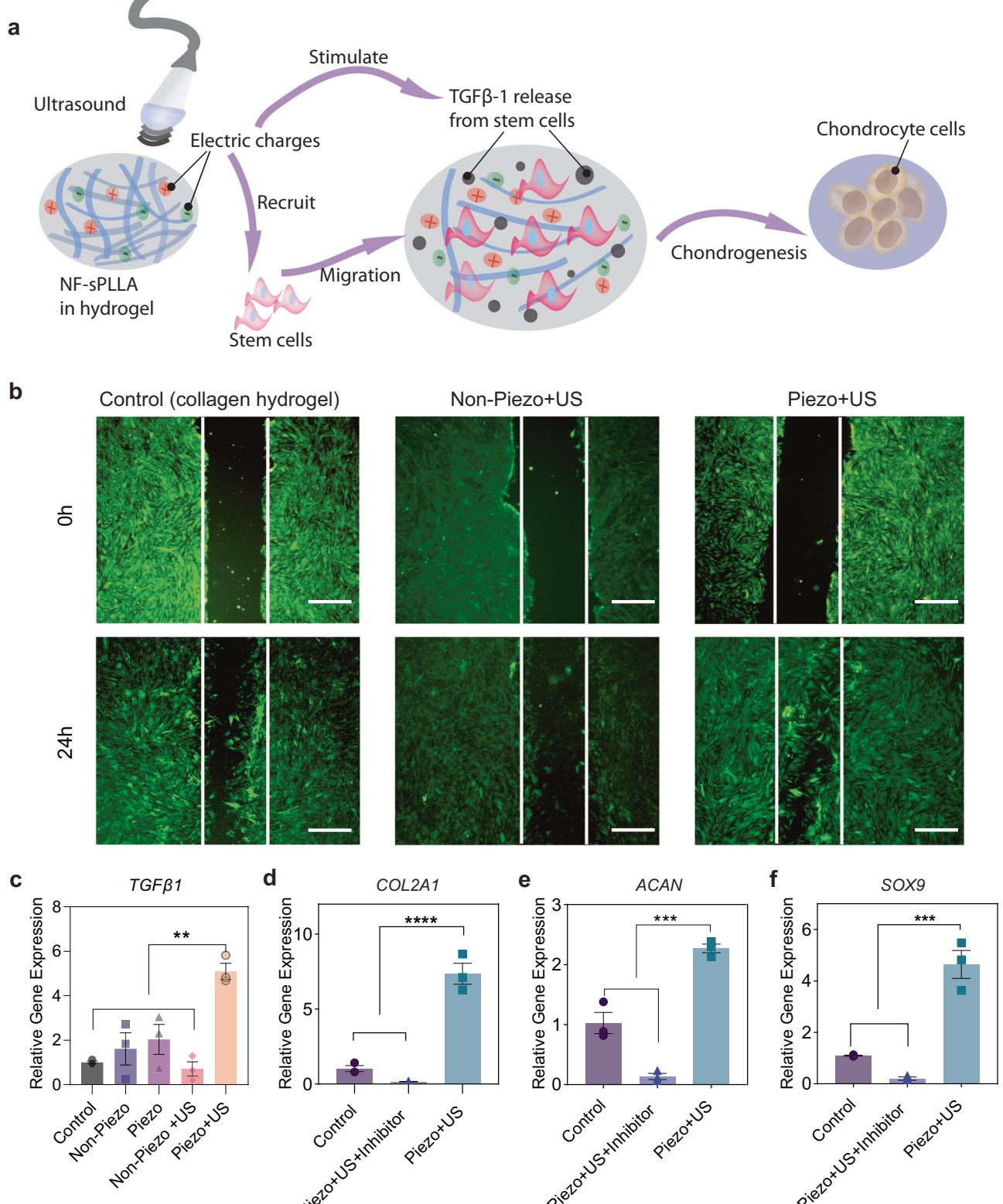

**Fig. 3 | Activated piezoelectricity induces chondrogenesis by recruiting the stem cells and stimulating the cells to secrete endogenous TGF-β1. a** Illustration of the hypothesis in which the piezoelectric hydrogel along with US activation recruits host cells and also induces the cells to release endogenous growth factors TGF-β1 which enhance cartilage healing. **b** Stem cell migration study evaluated by scratch test which was done by filling the wound bed with Piezo, Non-Piezo, and collagen hydrogel (Scale bars: 500 μm). **c** Relative gene *TGF-β1* expression after 2 days of culturing stem cells inside different hydrogels and stimulation conditions ($n = 3$ independent samples, the data are expressed as Mean ± SEM value, **$p < 0.01$, one-way ANOVA, Dunnett's multiple comparisons test). **d–f** Relative gene expression of the chondrogenic gene markers *COL2A1, ACAN*, and *SOX9* respectively from stem cells inside our US-activated piezoelectric hydrogel with and without TGF-β inhibitors ($n = 3$ independent samples, the data are expressed as mean ± SEM value ***$p < 0.001$, one-way ANOVA, Dunnett's multiple comparisons test). Exact $p$ values were provided in the Source Data file.

medium, the stem cells stopped expressing *COL2A1*, *SOX9*, and *ACAN* genes (Fig. 3d–f). This evidence once again strengthened our hypothesis that piezoelectric charge induces the chondrogenic differentiation of ADSCs by upregulating the expression of endogenous growth factor TGF-β1.

## Piezoelectric hydrogel induces cartilage healing in rabbit osteochondral (OC) defect model

After completing material characterization and the in vitro studies, we assessed the regenerative property of our piezoelectric hydrogel with US activation using a rabbit model with a critical-sized osteochondral defect (e.g., the cartilage defect that is large enough to avoid any self-healing). In this study, we drilled a critical-sized osteochondral defect into the trochlear groove of the knees for both hind legs (Supplementary Fig. 7a).

Before the cartilage healing study, we also conducted a test on rabbit cadavers' knee to confirm whether the same US condition used in vitro (frequency of 40 kHz and intensity of 0.33 watt/cm²) could penetrate through different tissues and reach the target site. We implanted three different sensors including a positive control PZT sensor, negative control polyimide, and experimental sensor fabricated from the PLLA nanofiber mat (similar to the sensor in a previous report[45]) into the trochlear groove defects. After closing the wound, the US was applied. Supplementary Fig. 7b showed that there were piezoelectric responses (output voltages) from PZT (positive control) and PLLA sensors, while the polyimide sensor only showed noise (for detailed discussion, see Supplementary Discussion). This indicates the US can access the cartilage defects where the PZT/PLLA sensors were implanted. We also provided details on safety considerations and treating dosage of our piezoelectric stimulation for the in vivo study in the Supplementary Discussion.

For this rabbit model, the critical-size defect was about 4 mm in diameter and 2 mm in depth. We had a total of 24 rabbits randomly divided into 6 groups (defect only, Non-Piezo, Piezo, defect + US, Non-Piezo + US, and experimental group Piezo + US) and 2 time points (1 month and 2 months of US activation/treatment). For each condition/group, there were 4 legs (*n* = 4). On the surgery day, when the defect was created, we injected 30 μl of different hydrogels into the defects and waited for 5 minutes to allow the hydrogel to solidify before suturing to close up the wound. The animals received the same US parameters in the in vitro study which was 40 kHz, 0.33 watt/cm² and 20 mins US treatment per day, 5 days per week for a 1- or 2-month period. The setup of the treatment is described in Supplementary Fig. 7a and Supplementary Movie 3. No side effects from the treatment were found during and after the treatment as the rabbits were able to walk around and behaved normally (Supplementary Movie 4).

After 1–2 months of treatment, the knees were harvested and analyzed with different methods. These methods include macroscopic evaluation, histological evaluation, mechanical characterization, and new subchondral bone quantification.

From the collected knees, we observed less cartilage and subchondral bone tissue in the defects of control and sham groups (either without applied US or without piezoelectric effect) compared to the experimental group of Piezo + US (Fig. 4a, b). Overall, the experimental group had a better macroscopic appearance with the new tissue integrating well with the defect border and having a higher degree of defect repair. The reformed tissue in Piezo + US after 2 months of treatment exhibited glossy white color cartilage, which resembled more of the surrounding native host tissues. In addition, we utilized ICRS criteria (Supplementary Table 1) to evaluate the healthiness of the knee[101]. As shown in Fig. 4c, there is a significant healing improvement in the gross view assessment on the US + piezoelectric hydrogel both for 1 month and 2 months of treatments, compared to other control/sham groups.

In the joint, subchondral bone provides support and elasticity that reduce impact shock on the cartilages during joint-loading[102]. Micro-computed tomography (μCT images and bone volume data revealed that after 1 month of treatment, the new subchondral bone formation rate was similar for all the knees in different groups. However, after 2 months, the amount of subchondral bone found inside the defects of Piezo + US knee joints was significantly higher than those of other groups and the 1-month time points (Fig. 4b, d).

For histological evaluation, we employed (1) Safranin O/fast green staining to visualize hyaline cartilage/sulfated GAGs. (2) Collagen II staining to study collagen formation and collagen X for hypertrophic chondrocyte. (3) Hematoxylin and Eosin (H&E) for tissue morphology evaluation, inflammatory infiltration, and cell apoptosis. As shown in Fig. 5a, after 1–2 months of piezoelectric treatment, more articular cartilage was found inside the defects of the Piezo + US group. Noticeably, the newly formed cartilage tissues (black arrows) in the experimental group of piezo hydrogel + US were well-integrated with the native host tissue (yellow markers), with a clear chondrocyte structure and cell distribution. Furthermore, in the experimental group (Piezo + US) samples at 2-month time point, as shown in Supplementary Fig. 7c, d, the stained cells showed a clear organization and structure of hyaline cartilage. There is a clear superficial zone with a high number of flattened chondrocyte cells (black arrows), an intermediate zone with spherical chondrocytes shape (violet arrows), a deep zone with columnar orientated chondrocytes (yellow arrows) and a tidemark (white dash lines). In contrast, the sham and control groups mainly formed fibrosis scar tissue (hot pink arrows) or bony tissue (red arrows). The reformed tissues inside the cartilage defects of the control/sham groups were sometimes detached from native tissue (violet markers), a phenomenon, which is usually observed in OA[103]. Although the histology image (Fig. 5a) indicates the Non-Piezo without US group at 2 months had more tissue filling in the defect than the 1-month result, both images at 1 and 2 months show that the defects were filtrated with fibrosis scar tissue (hot pink arrow). Additionally, the fibrosis tissue in the Non-piezo - US group at 2 months was detached from native tissue, similarly to the 1-month data (violet markers). Hence, Non-Piezo samples without US at 1- or 2-month time point are almost the same in terms of cartilage regeneration. Collagen II staining reveals that the experimental groups (Piezo + US) exhibited highly positive collagen II staining, indicating abundant collagen II protein within the defects. More importantly, the Piezo + US group demonstrated better collagen fibril structure, similar to native tissues after 2 months via the immune-histological staining. In contrast, the other groups showed minimal to no observable collagen II production. Furthermore, Supplementary Fig. 8 illustrates hypertrophic chondrocyte cells with collagen X staining. The data demonstrates that; for the Piezo + US group, both at 1- and 2-month time points, the newly formed tissues mostly appeared in the background color (lavender), indicating the absence of collagen X (dark brown). Noticeably, the Piezo group without US showed non-collagen X tissue at the 1-month time point, but after 2 months, the tissue transformed into hypertrophic cartilage. Meanwhile, the other groups showed highly positive collagen X staining in the newly formed tissues both at 1- and 2-month time points. In the H&E staining, there was no inflammation or cell apoptosis observed in all the groups which indicates that Piezo and Non-piezo hydrogel with and without US are non-toxic to the animal (Fig. 5a and Supplementary Fig. 9). Next, we assessed cartilage regeneration based on ICRS histological evaluations. ICRS (Supplementary Table 2) histological evaluation indicated that Piezo + US samples had significantly higher scores due to the better smoothness in the surfaces, more hyaline cartilage matrix regeneration, and higher cell viability and cell distribution (Fig. 5b).

For mechanical properties, we employed nanoindentation to assess the reduced modulus of the newly generated cartilage tissue. Fig. 5c shows reduced moduli of different cartilage samples including

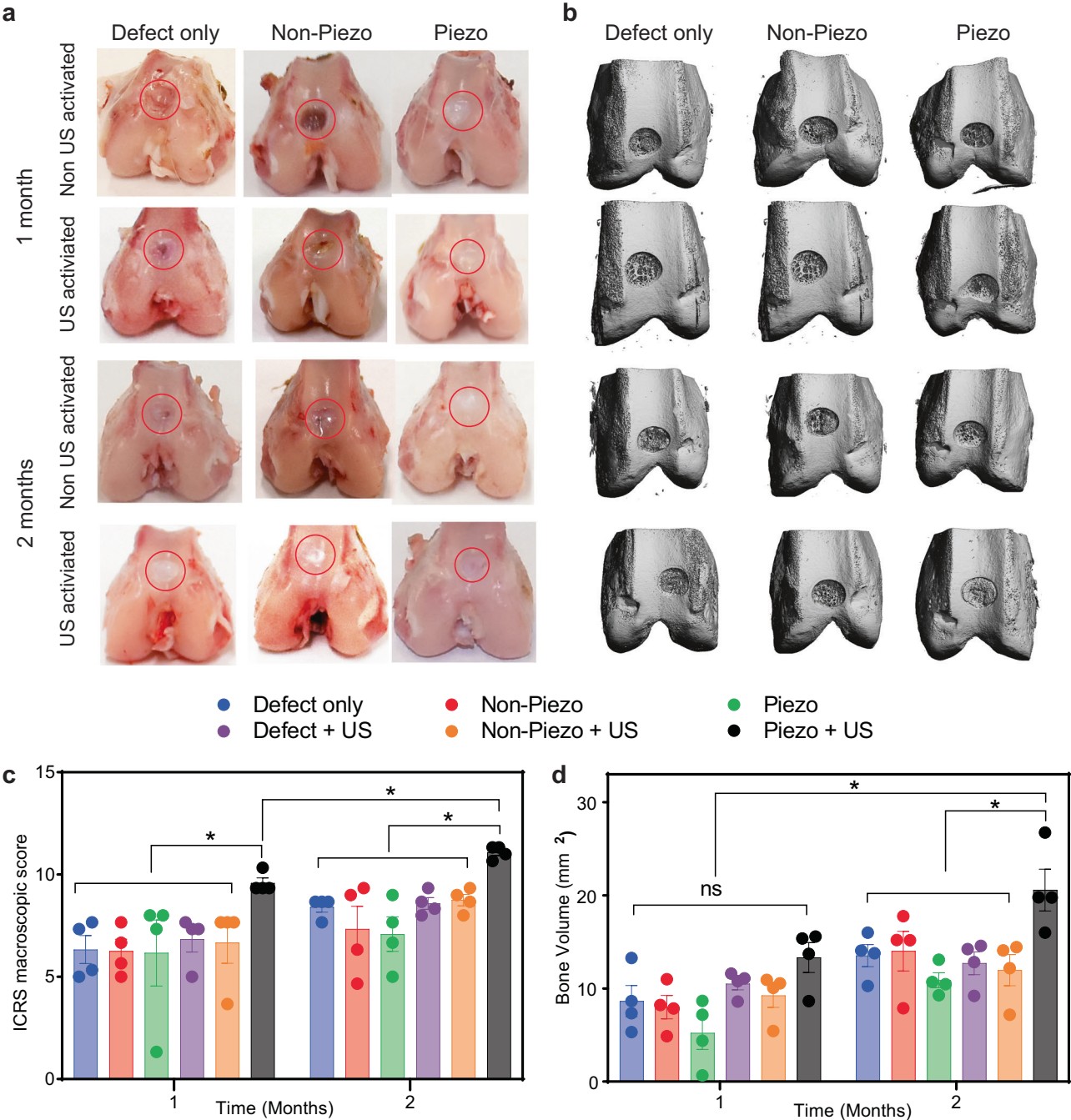

**Fig. 4 | Piezoelectric hydrogel enhances cartilage healing evaluated by macroscopic scoring, and subchondral bone formation in vivo. a** The digital photographs showing the rabbit femurs with defect only, non-piezo, and piezo hydrogel group with or without US activation (1–2 months). The red circle marks where the defect was originally created. **b** Reconstruction of the bone on femurs using μ-CT. **c** ICRS score for macroscopic cartilage evaluation for 1- and 2-months US activation on rabbit knees (*n* = 4 knees for each group, the score was an average point from three independent professionals and blinded evaluation, the data are

expressed as data points with Mean ± SEM, *p < 0.05, one-way ANOVA, Two-stage linear step-up procedure of Benjamini, Krieger and Yekutieli were use for the sample in the same time point. *t* test, two-tailed was used to compare Piezo + US (1 month) vs Piezo + US (2 months)). **d** Volume of subchondral bone formed inside defect after 1 or 2 months of US treatment (*n* = 4 knees for each group, the data are expressed as points with Mean ± SEM. *p < 0.05, n.s = not significant, one-way ANOVA, Dunnett's multiple comparisons test). Exact *p* value were provided in the Source Data file.

normal hyaline cartilage of healthy tissues. A significantly higher modulus value was achieved from the experimental group (piezo + US) from both 1 and 2 months of treatment compared to other sham and control groups. The modulus value after 2 months of the piezoelectric treatment (-5.3 ± 0.3 GPa) is approaching that of the healthy native cartilages (6.04 ± 0.4 GPa). This can be also seen in the load-displacement curve of the 2-month piezo + US group which gets closer to that of the native healthy cartilage (Fig. 5d).

Collectively, we have shown that the injectable and biodegradable piezoelectric hydrogel activated by the external US can induce chondrogenesis in vitro and treat severe OC defects in vivo. The piezoelectric hydrogel with US activation upregulates chondrogenic genes including *COL2A1*, *ACAN*, and *SOX9* from cultured stem cells, leading to the increase of GAG, and collagen type 2 production in vitro. Furthermore, the piezoelectric hydrogel under US activation could reduce inflammation by decreasing TNF-alpha produced by macrophages.

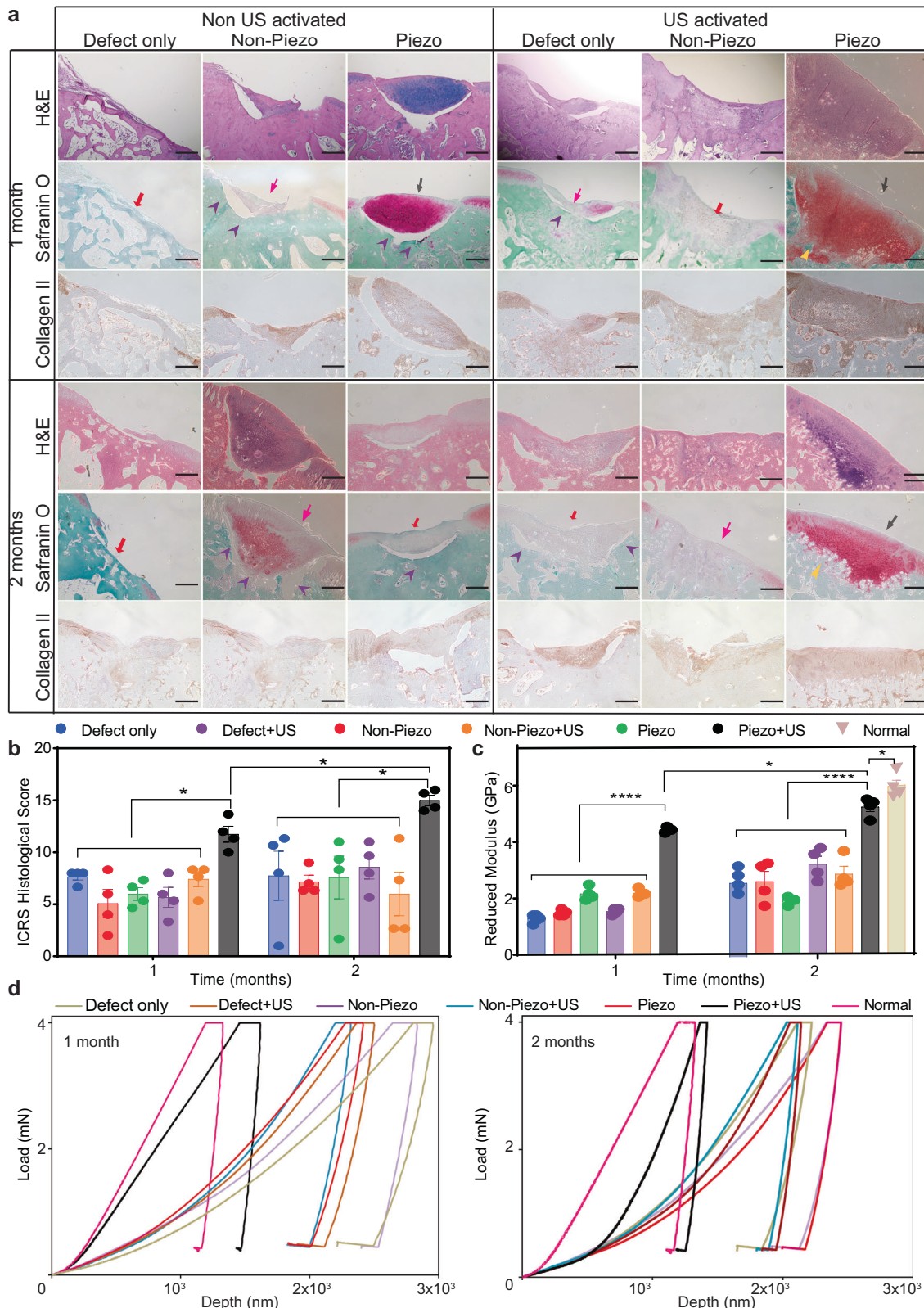

In vivo data demonstrates that the piezoelectric hydrogel with US activation induces cartilage healing in the rabbit critical-size OC defect model. After 2 months of the piezoelectric treatment (Piezo + US), there is a significantly higher amount of subchondral bone formation, clear and organized articular cartilage structure, and better cartilage mechanical properties in the experimental group (Piezo + US) compared to the other control and sham groups. The mechanistic study suggests that the regenerative process could be started by the US-activated piezoelectric hydrogel to recruit host cells, and then induce the cells to secrete TGF-β cytokine, leading to the chondrogenic differentiation via the autocrine pathway. Finally, these chondrocyte cells deposited cartilage matrix to regenerate the lost tissue.

In previous research, we succeeded in repairing the osteochondral defect model by implanting solid biodegradable piezoelectric

**Fig. 5 | Piezoelectric hydrogel induces cartilage healing, evaluated by histology assessment and mechanical testing in vivo. a** H&E staining and Safranin O/fast green and collagen II staining to evaluate the articular cartilage regeneration for sham (defect only), non-piezo/piezo PLLA hydrogels with and without US activation (1–2 months). Black arrows indicate newly formed cartilage tissues. Yellow markers indicate the new cartilage tissue which was well-integrated with the native host tissue. Hot pink arrows indicate fibrillation filling, red arrows indicate bony tissue and violet markers indicate the detachment of newly formed tissue from the host ($n$ = 4 knees for each group, scale bars: 500 μm). **b** ICRS histological evaluation, ($n$ = 4 knees for each group, the score was an average point from three independent professionals and a blinded evaluation, the data are expressed as data points with Mean ± SEM *$p$ < 0.05, one-way ANOVA, Dunnett's multiple comparisons test were

used for the sample in the same time point. t-test, two-tailed was used to compare Piezo + US (1 month) vs Piezo + US (2 months)). **c** Reduced modulus of newly formed cartilage inside the defect of different hydrogel groups with and without 1–2 month US activation. Normal healthy rabbit cartilage serves as a positive control ($n$ = 4 knees for each group, random indentation-testing position, the data are expressed as boxes with points and means ± SEM. ****$p$ < 0.0001, *$p$ < 0.05, one-way ANOVA Dunnett's multiple comparisons test were used for the sample in the same time point. t-test, two-tailed was used to compare Piezo + US (1 month) vs Piezo + US (2 months)). **d** Representative indentation curves for different groups indicate 1 month of treatment (left) and 2 months of treatment (right). Exact $p$ value were provided in the Source Data file.

scaffolds into the knee joints and inducing joint load by physical exercise[44]. Here, we provide an alternative approach to healing OC defects without involving major surgery. It's important to note that the locations of the cartilage defects were different between the two studies. In this study, osteochondral defects were created in the trochlear groove, and in the previous study, the defects were in the femoral condyle. We chose not to use the femoral condyle defect because the piezoelectric hydrogel could be activated by the animal movements, which would interfere with the results. Therefore, it is challenging to establish an equitable comparison between the two studies since the intrinsic generation rates of these two locations may be different. However, our reported biodegradable piezoelectric scaffold requires an invasive knee-opened surgery for implantation, which is associated with risks of many complications (infections, arthrofibrosis, increasing pain, postoperative swelling, and the need for longer recovery time)[104]. Herein, we provided a significantly less invasive method with our injectable and biodegradable piezoelectric hydrogel to heal osteochondral defects. The fabricated hydrogel can be delivered to the wound site by a simple and minimally invasive injection that allows us to avoid open-knee implantation, therefore, reducing the associated complications.

There are still limitations and challenges to overcome and improve the regenerative properties of this piezoelectric hydrogel. First, in the in vitro study, we noticed that different trends of chondrogenic marker genes (*COL2A1*, *ACAN*, and *SOX9*) were observed in Non-Piezo, Piezo, Non-Piezo + US groups. Consequently, additional research could be considered to provide a deep understanding of this phenomenon. Second, for the animal model, we used rabbits in this study, which have a much thinner cartilage layer (250 – 700 μm) and less body weight compared to humans (e.g. cartilage layer of 1–3 mm). Therefore, a study with a large animal model such as sheep or horses which have a similar cartilage thickness and body weight to the human should be performed. Third, US parameters may need to be re-evaluated for larger animal or human studies. Because larger animals or humans may have different anatomy and physiology, including variations in skin, ligament, and muscle thickness, compared to rabbits. Therefore, adjustment of ultrasound parameters may be needed to penetrate through these tissues to reach the defect and effectively trigger piezoelectricity of NF-sPLLA fibers. Fortunately, this should be easy to accomplish, based on the development of ultrasound therapies which have already been widely used for various medical applications in humans[105] including joint pain management[106,107] and bone healing (e.g.:AccelStim)[108]. Also, the sample size of the animals used in this study is relatively small ($n$ = 4)[109–111]. Hence, increasing the sample size is necessary in the next phase of this project. Fourth, hyaline cartilage has heterogeneous structures including the superficial zone, transitional zone, and deep zone with increasing GAG content distribution (Supplementary Fig. 7c). Thus, in a future study, the NF-sPLLA concentration can be tuned at different hydrogel layers to match the property of each cartilage layer. Our data in Fig. 2g–i supports such a tunability by indicating that different amounts of PLLA fibers inside the collagen hydrogel result in different *ACAN* and collagen type II mRNA

expressions. Finally, this study was done over a short period (2-3 months) as a proof of concept to demonstrate that piezoelectric hydrogel + US can exhibit effective cartilage regeneration. However, the mechanical properties of the regenerated tissues at the end point of 2 months of the stimulation still did not match exactly with the native normal cartilage (undamaged cartilage tissue). As such, a longer period of treatment may be needed for the tissue to perfectly regenerate. A future study with a longer endpoint is essential to elucidate this point.

Regardless of these future works, we have presented a piezoelectric hydrogel that can (1) be injected into the body via a minimally invasive process to preclude implantation surgery, (2) self-generate electrical cues to promote cartilage and potentially heal other tissues under US activation, and (3) eventually, degrade into safe degradation byproducts to avoid invasive removal surgery and any harm to the body. These three significant advantages have not been achieved by any other reported hydrogel for tissue regeneration. The presented injectable piezoelectric hydrogel along with the US activation approach is potentially applicable to the regeneration of other damaged tissues such as bone, nerve, muscle, skin, etc. Furthermore, this piezoelectric hydrogel can be used as a platform to create other injectable devices such as actuators, sensors, energy harvesters, and transducers for other medical applications, thus, offering a significant impact not only on the field of tissue regenerative engineering but potentially other areas such as drug delivery, health monitoring, and other disease treatments/preventions.

## Methods
### Preparation of injectable piezoelectric PLLA nanofibers hydrogel
First, piezoelectric PLLA nanofiber and non-piezoelectric fibers PDLLA mats were prepared by the electrospinning method[55]. 1 g of PLLA or PDLLA (Corbion Purac Amsterdam, Netherlands) was dissolved in 20 ml of Chloroform (Sigma) overnight. The solutions were spun at 2 ml/h flow rate using G22 needle under 14 kV and collected on a 4000 rpm drum to achieve aligned nanofiber mats and the humidity was controlled ~25–45% (Supplementary Fig. 1). PLLA mats were then annealed at 105 °C overnight and allowed to slowly cool to room temperature before repeating the process at 160.1 °C.

To fabricate NF-sPLLA and NF-sPDLLA, PLLA and PDLLA mats were placed in Peel-A-Way molds (Fisher Scientific) with cryostat embedding medium O.C.T. compound and stored at −80 °C for 2 hours to achieve solidified blocks. These blocks were sectioned perpendicular with fiber direction to make a 25 μm length using Leica Cryostat. We then collected the sections and cleaned them with DI water until the fibers were cleaned (confirmed by NMR).

Before use, piezoelectric and non-piezoelectric nanofibers were mixed with collagen I rat tail (A1048301, Gibco) naturalized with sodium hydroxide 1 N and 10× PBS according to the manufacturer's recommendation to make them into injectable hydrogels.

For the material characterizations such as SEM, wettability, and piezoelectric property evaluation, 5 mg/ml of piezoelectric PLLA and

non-piezoelectric PDLLA hydrogels were vacuumed dry overnight to create dried scaffolds. The dried scaffolds were only used for material characterization purposes whereas wet scaffolds were not feasible to use.

### Measurement of open-circuit output voltage under ultrasound system

The piezo and non-piezo-dried scaffolds were then cut into $1 \times 1$ cm squares. To measure the piezoelectric property, 2 electrodes made from heavy-duty aluminum ($1 \times 1$ cm) were sandwiched with scaffolds, and polyimide tape (DuPont) was used to encapsulate the electrodes. The exposed Al foil electrode leads were then reinforced using copper tape (Ted Pella). These sensors were subjected to ultrasonic waves at 1 MHz.

It is noteworthy that we utilized 10×PBS solution and NaOH to crosslink collagen hydrogel, making the liquid composite hydrogel conductive due to the high concentration of salts. Because of the conductivity properties of the composite hydrogel in its liquid form, it is not feasible to directly measure the piezoelectric output of these hydrogels in their original wet form. Hence, we adopted a vacuum drying method to obtain dried forms of the composite hydrogels, which enabled accurate measurement of the piezoelectric output. This approach was demonstrated in our previous publication, where dried scaffolds were utilized to measure the output voltage of 3D piezoelectric scaffolds[44].

### Measurement of open-circuit output voltage of different NF-sPLLA dried hydrogel concentrations under ultrasound system

Different concentration (1, 5, or 10 mg/ml) NF-sPLLA hydrogels (vacuumed dry) were cut into $1 \times 1$ cm squares. To measure the piezoelectric property, 2 electrodes made from heavy-duty aluminum (Al) ($1 \times 1$ cm) were sandwiched with scaffolds, and polyimide tape (DuPont) was used to encapsulate the electrodes. The exposed Al foil electrode leads were then reinforced using copper tape (Ted Pella). These sensors were subjected to ultrasonic waves at 40 KHz and 0.33 watt/cm². It should be noted that due to the high electromagnetic interference noise at 40 KHz, we further validated the results by measuring the piezoelectric voltage output at 1 MHz.

### Material characterizations

**Scanning electron microscopy (SEM).** SEM was used to image the PLLA film, short fibers, and dried scaffold. The microstructure, morphology, and porosity of the dried scaffold were observed. PLLA film, short fibers, and dried scaffolds were mounted on a standard SEM pin (Ted Pella, Redding, CA) using carbon conductive tape (Ted Pella, Redding, CA). The samples were coated in gold-palladium with a sputter coater and then imaged with a FEI TeneoLoVac SEM at 5 kV and 5000x magnification.

**X-ray Diffraction (XRD) and Differential Scanning Calorimetry (DSC).** All sample measurements were carried out at RT using a Bruker D2 Phaser, with 0.1 °C/min scanning speed, from 5° to 45 °C. DSC analysis was performed on annealed samples using a DSC Q-100. Samples (3–5 mg) were packed inside aluminum pans and closed with lids (PerkinElmer), and empty sealed pans were used as references. The samples were heated from 0 °C to 210 °C in the nitrogen atmosphere with 10 °C/min ramping rate.

**Hemolysis test.** Healthy rabbit whole blood (with Na citrate anticoagulant, from Innovative Research) was diluted with normal saline at 4:5 volume ratio. 300 µl of piezoelectric and non-piezoelectric hydrogels were incubated at 37 °C for 30 mins for hydrogels to solidify[44]. These scaffolds were then incubated in 10 ml of normal saline with 200 µl of diluted blood for 60 mins at 37 °C. Saline and DI water without scaffolds were used as the negative and positive control

respectively. These samples were centrifuged at 1000 g for 5 mins, the supernatant was collected to a 96 well plate, and absorbance at 545 nm was carried out. The percentage of hemolysis was calculated as

$$\text{Hemolysis rate (\%)} = \frac{(\text{testing sample} - \text{negative control})}{(\text{positive control} - \text{negative control})} \times 100\% \tag{1}$$

**Swelling measurement.** In all, 5 mg/ml of NF-sPLLA were mixed with collagen I rat tail (A1048301, Gibco) naturalized with sodium hydroxide 1 N and 10× PBS according to the manufacturer's recommendation. Collagen hydrogel was prepared in a similar manner. 300 µl of each hydrogel was added to a 24 well plate and incubated at 37 °C for gelation. The solidified hydrogels were vacuumed dry overnight to generate dried scaffolds. For the swelling test, dry scaffolds were weighed ($M_o$) immersed in 1× PBS (pH = 7.4) and gently shaken at RT. At each time point, the sample's weight was recorded ($M_t$), and the swelling ratio was calculated as:

$$\text{SW} = \frac{Mt - Mo}{Mo} \times 100\% \tag{2}$$

**Nuclear Magnetic Resonance spectroscopy (NMR).** To confirm PLAA short fibers were solvent-free, residual solvent analysis was employed using a Bruker BMX 500 MHz High-Resolution NMR spectrometer. 10 mg PLLA short fibers were dissolved in 1 ml of deuterated DCM (Millipore Sigma, Burlington, MA). 500 µl of solution was added to an NMR tube (Wilmad 500 Mhz, Millipore Sigma, Burlington, MA). The sample was run with the $^1$H NMR test and 500ppm was set as the detection limit. The recorded data was processed using Mnova 14.2.0.

**Rheological testing.** Rheological characterizations of different samples were carried out through AR-G2 rheometer (TA instruments) using a 20 mm diameter plate and the height gap was set at 500 µm. To evaluate hydrogel properties, 500 µl of sample was pipetted onto a rheometer plate, the geometry was lowered slowly until it contacted the hydrogel. We then gently removed excess hydrogel and prevented the hydrogel from drying by using moisture traps.

First, strain sweeps were performed, after loading sample to the plate, the plate was set at 37 °C. After 30 minutes, when the sample reached equilibrium, the tests were recorded from 0.01 % to 100 % strain at 1 Hz. The results indicated both samples were stable when 0.01% to 10% strain was applied. Therefore, in the frequency sweeps, we recorded data of 0.01 to 100 Hz frequency with 10% stain applied over 30 mins for equilibrium at 37 °C[112]. To study hydrogel solidification time, strain was set at 10% and 1 Hz frequency. The viscosity test was measured with a range of shear rate from 0.1–50 rad s$^{-1}$ [62].

**Inject hydrogel with X-RAY guidance.** A rabbit cadaver was obtained from Animal Technologies (TX, USA). Before surgery, the animal leg was shaved and disinfected. After that the knee joint of the rabbits was revealed by medial parapatellar arthrotomy, and the patella was displaced. In the trochlear groove of the femur, a critical-size osteochondral defect was drilled with a 4 mm diameter and 2 mm depth. Once the defect was created and cleaned well with saline, the muscle was closed by interrupted sutures and finally the leg skin was closed by intradermal sutures. To view the injection under x-ray, the piezoelectric hydrogel was mixed with Optiray (1:1 v/v ratio). The defect was then identified under a C-Arm Fluoroscope, through x-ray guidance, and hydrogel was loaded into the defect through a G24 needle.

**Degradation study.** In all, 5 mg/ml of NF-sPLLA were mixed with collagen I rat tail (A1048301, Gibco) naturalized with sodium hydroxide 1 N and 10× PBS according to the manufacturer's recommendation.

300 μl of each hydrogel was added to 24 well plate and incubated at 37 °C for gelation. After that, these scaffolds were divided into four groups to test degradation in media, PBS with and without US stimulation at 37 °C for 2 months, and then accelerated condition at 80 °C for 30 mins. Photographs of the scaffold were taken.

## Cell culture

Rabbit ADSCs (RBXMD-01001, Cyagen), Human THP-1 (ATCC) were used for in vitro testing. The cells were cultured in growth medium according to the vendor's guidance. In brief, cells were grown in α-MEM supplemented with 10% FBS, 1% penicillin and streptomycin, 1% l-glutamine, 0.2 mM ascorbic acid, and 1 ng/mg basic fibroblast growth factor. Cells at passage 3 or 4 were used for further experiments and the medium was refreshed every 2-3 days.

## Viability assay

In all, $10^5$ ADSCs were mixed with 300 μl of nanofiber hydrogels. PrestoBlue (A13261, Thermo Fisher) was used to determine the viability of cells at different time points.

## Migration test

The cells were cultured in glass bottom dishes (150682, Thermo Fisher) until confluent. A 200 μl tip was used to make a scratch on the monolayer of cells, the medium was removed, and the cells were washed three times with 1× PBS to remove the floating cells. After that, 200 μl of hydrogels with or without fibers were added and allowed to solidify for 20 mins before new media was added. 24 hours later, the cells were stained with live-cell fluorescence.

## Chondrogenesis evaluation

ADSCs were mixed with hydrogels with/without fibers to create $5 \times 10^6$ cells/ml in 24 well plates. In these tests, we used both growth medium and chondrogenic medium. The formulation of the growth medium is the same medium that was used for cell expansion. For chondrogenic medium, the formulation included DMEM supplemented with 100 μg/ml sodium pyruvate, 0.2 mM ascorbic acid 2-phosphate, 1% penicillin and streptomycin, 50 μg/ml Insulin-Transferrin-Selenite premix (ITS). Before use, 0.1 μM dexamethasone and 10 ng/ml TGF-β3 were freshly added. In the experiment addressing the effect of TGF-β on chondrogenesis, we used normal growth medium without TGF-β and then added 1 μM TGF-β inhibitor (SB431542, Sigma-Aldrich). The growth medium contains α-MEM supplemented with 1% penicillin and streptomycin, 1% l-glutamine, 0.2 mM ascorbic acid, and 1 ng/mg basic fibroblast growth factor.

## Ultrasound (US) induced piezoelectric stimulation on the ADSCs

For in vitro treatment, an ultrasonic bath (Branson 2800 CPX series) was employed which provided 40 Khz US stimulation with 0.33 Watt cm$^{-2}$. Cells started getting treatments after 24 hours of seeding on different hydrogels, 20 mins per day, for the 14-day period including weekends. To prevent contamination, the bath was disinfected with 70% ethanol and refilled with distilled water every day. Also, the well plates were sealed with a layer of plastic wrap and 2 layers of duct tape before we placed them into the bath. To make sure the well plates were in the center of the bath, a laboratory clamp holder was used.

## RNA isolation, quantitative polymerase chain reaction (PCR)

For RNA collection and rt-qPCR performance, the cells were collected after 14 days of treatment. First, RNA was isolated using TRIzol™ Plus RNA Purification Kit (12183555, Thermos Fisher) according to the vendor's protocol. The RNA was transformed to cDNA using an iScript Synthesis kit (1708897, Bio-Rad). After that, real-time quantitative PCR was performed with different genes of interest including *B2M*, type II

Collagen (*COL2A1*), *SOX9*, Agrecan (*ACAN*), *TGF-β1, Beta Actin,* and *TNF alpha*. All the primers were used as follow:

*B2M*, Forward: 5′TGA AAC ATG TCA CTC TCA AG 3′ and Reverse: 5′ AGA CAC AAA TGT TAG CCT TC 3′

*COL2A1*, Forward: 5′TTC TCC TTT CTG CCC CTT TGG T 3′ and Reverse: 5′TCT GTG AAG ACA CCA AGG ACT G 3′

*SOX9*, Forward: 5′CTC CGA CAC CGA GAA TAC A 3′ and Reverse: 5′ CCT CTT CGC TCT CCT TCT T 3′

*ACAN*, Forward: 5′CAG CCG GAC AAC TTC TTT 3′ and Reverse: 5′ GTG AAG GGT AGG TGG TAA TTG 3′

*TGF-β1*, Forward: 5′CTG GGT CAC TCC CAA ATA 3′ and Reverse: 5′ ACA AGC AGG TGG AAG AT 3′

*Beta Actin*, Forward: 5′GAG GTA TCC TGA CCC TGA AGT A 3′ and Reverse: 5′CAC ACG CAG CTC ATT GTA GA 3′

*TNF alpha*, Forward: 5′GAT CCC TGA CAT CTG GAA TCT G 3′ and Reverse: GAA ACA TCT GGA GAG AGG AAG G 3′

B2M beta-2 microglobulin gene was used as a housekeeping gene. The relative gene expression data was calculated based on $2^{-\Delta\Delta Ct}$ method.

## GAG assay

For GAG quantification, at day 14 of cultivation, each sample of cell-pellet was digested with 250 μl of tris-HCl buffer with 0.05 M tris and 1 mM $CaCl_2$ at pH 8.0 mixed with 1 mg/ml proteinase K at 56 °C for 16 hours. 25 μl of samples were then mixed with 150 μl of 1,9-dimethyl methylene blue (DMMB) solution that contained 3.04 g/l glycine, 2.38 g/l sodium chloride, and 20 mg/l DMMB in DI water. The data was carried out by absorbance reading at 525 nm wavelength using a microplate reader.

## Alcian blue staining and immunofluorescent staining

At 14 days, samples were first washed three times with 1× PBS and fixed in 4% formalin for 3 hours. Scaffolds were then embedded in paraffin wax and sectioned into 5 μm thickness. The sides were then deparaffinized with xylene 2 times and rehydrated with a series of ethanol from 50%, 75%, 95% to 100%.

For Alcian blue staining, those slides were immersed with Alcian blue solution for 30 mins and then counterstained with nuclear fast red for 5 mins.

For collagen type II immunofluorescent staining, the samples were incubated with proteinase K (5 μg/ml) at room temperature for 10 mins and an additional 40 mins with hyaluronidase (1 mg/ml at) at 37 °C. To block the unspecific binding of antibodies, the samples were incubated with 5% BSA for 60 mins. After that, samples were incubated with an antibody (AF5710, OriGene) in a dark chamber overnight at 4 °C. The next day, slides were carefully rinsed three times with PBST and then incubated with Alexa Fluor 488 labeled goat anti-mouse IgG (H + L) secondary antibody for 2 hours at room temperature. Slides were washed with PBS and mounted with a medium containing DAPI (4′,6-diamidino-2-phenylindole).

## Validating the penetration of 40 KHz to rabbit knee and activating piezoelectric sensors

Sensors made of PZT, PLLA, and polyimide were fabricated following the same process as described above but with a smaller size of 5 × 5 mm. In a rabbit cadaver's knee, an incision was made to create the defect, and subsequently, the sensor part of the PZT, PLLA, and polyimide sensors were inserted individually into the defect. The tail sensor was then secured with ligaments using sutures. Finally, the muscle and skin were closed.

To measure the output voltage of these sensors, a 40 KHz ultrasound (similar to Supplementary Movie 3) was placed on top of the knee. The data acquisition was performed using PicoScope software (version 6), and the signals were collected and analyzed using MATLAB.

## Osteochondral defect surgical procedure

The study followed the guidelines and recommendations of the Guide for the Care and Use of Laboratory Animals of the National Institutes of Health. All procedures were approved by IACUC, University of Connecticut (#TE-102090-0622 and AP-200653). 24 male *New Zealand* white rabbits (weighing ~3 kg) were randomly assigned to six groups (12 rabbits for 1 month and another 12 rabbits for 2 months of treatment) for in vivo experiments. Osteochondral defect surgery was performed on each knee of the animals[44]. In detail, the rabbit was given a half dose of buprenorphine (0.02–0.05 mg/kg) as a preoperative analgesic. After that, each rabbit was anesthetized with a mixture of ketamine/xylazine/atropine and maintained with 2.5–3.5% isoflurane. The knee joint of the rabbits was revealed by medial parapatellar arthrotomy, and the patella was displaced. In the trochlear groove of the femur, a critical-size osteochondral defect was drilled with a 4 mm diameter and 2 mm depth. The piezoelectric or non-piezoelectric hydrogels were then filled inside the defect. 5 mins later when the hydrogel was solidified inside the wound bed, the joint was closed with sutures. After surgery, animals were under observation until they became sternly recumbent after which they were immediately given the other half of the buprenorphine dose and then returned to housing.

## US rabbit treatments

After 3 weeks of surgery, the animals received US treatment 20 mins per day, 5 days per week for 1-month or 2-month time points. To maintain consistency between the in vivo and in vitro experiments, we applied the same parameters that were used in vitro to in vivo. A 40 kHz US generator (Steminc) equipped with 2 of 40 KHz bolt clamped Langevin transducers (Steminc) connected in series were used in the experiment. This setup allowed us to generate an intensity of 0.33 Watt cm$^{-2}$. First, the animals were anesthetized with acepromazine maleate (0.1 ml/kg) and maintained with 3% isoflurane. After shaving, the rabbit's knees were kept stable with a laboratory clamp, a layer of ultrasound gel (Aquasonic) was applied, and a transducer was gently placed on top of the knee joint (Supplementary Movie 3).

## Micro-CT

After 1 or 2 months of treatments, rabbit's knees were collected and fixed with 10% neutral-buffered formalin. The knee was then placed inside a sample tube before micro-CT scanning (μCT40, Scanco Medical, 55 kVp, 145 μA, 300 ms integration). The resolution was set at 20 μm/voxel and 1024 × 1024 pixels. 3D geometry reconstruction was performed with Dragonfly 2020.1.1.809 (Object Research Systems). The new bone formation volume was calculated by 3D Slicer 5.2.1 software[113].

## Histology and IHC of collagen II, Collagen X

After micro-CT scanning, samples were then dehydrated in graded alcohol, decalcified. Before sectioning, samples were embedded in wax. Finally, samples were sectioned for H&E, safranin O, and IHC staining ab34712 (Abcam) for collagen II and ab49945 (Abcam) for collagen X according to the vendor's protocol.

## Score system to evaluate the degree of cartilage regeneration

Independent professionals who have knowledge and experience with cartilage histology performed the morphological evaluation of cartilage regeneration independently. The evaluation was based on ICRS macroscopic evaluation of cartilage[101] and ICRS Visual Histological Assessment scale[114].

## Biomechanical testing

A nanoindentation machine (iNano, KLA) was used to evaluate the mechanical properties of the new form of cartilage. In this experiment, the diamond Berkovich was deployed and calibrated on the reference surface before each test. Based on other reports, we deployed 4 mN for Target and 0.2 s$^{-1}$ for Strain Rate as parameters for biomechanical testing[115,116]. Healthy rabbit knees were collected and processed in the same condition as other samples for positive control.

## Statistics and reproducibility

All data are plotted as data points or boxes with data points and expressed as Means ± SEM. The number of samples indicated specificity in each test. Statistical significance was calculated using one-way ANOVA or t-test where the test was appropriate with GraphPad Prism 9, exact *p* value were provided in Source Data file. The data presented using the photograph and micrograph was repeated 3 times (unless otherwise indicated) with similar results.

## Reporting summary

Further information on research design is available in the Nature Portfolio Reporting Summary linked to this article.

# Data availability

All data supporting the findings described in this manuscript are available within the paper, the Supplementary Information, and Source data file. The full image dataset is available from the corresponding author upon request. Source data are provided with this paper.

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

## Acknowledgements

The project was funded by the NIH (grant number R21AR074645 to T.D.N. and partially by grant numbers R21AR078744, R21AR080919, and R01AR080102 to T.D.N.). We thank the Center for Comparative Medicine and Histology Core, especially Dr. Zhifang Hao University of Connecticut Health Center for their hard work in supporting us in this study. We thank Jordan Vinkoor for his assistance with editing this manuscript.

## Author contributions

T.D.N. conceived the idea and designed the project. T.D.N. and T.V. designed the experiments. T.D.N, T.V, and T.T.L. wrote the manuscript. E.D., S.P., S.T., M.T.C., and F.L., helped edit the manuscript. T.V. fabricated the material and hydrogels. T.T.L., E.J.C., and J.P. did sensor fabrication, and piezoelectric testing and analyzed the data. T.T.L. performed SEM and NMR studies. T.V., G.K.D., Y.L., H.K., L.S., S.K., R. M. C., performed surgeries, T.V., L.S., S.K., R. M. C., S.B., treated Ultrasound for animals. T.V., P.S., performed the in vitro cell study hemolysis study, and rheometer study. T.V., G.K.D, M.A.M, J.H.C collected tissues, and performed the injection study with X-Ray. E.R., H.W., and H.K., provided a professional evaluation of the in vivo results. T.V. did bone volume analysis. Q.W provided histology analysis and insights information for regeneration in vivo. T.V., S.Y., S.W.L. performed nanoindentation tests. R.M.C., K.W.H.L., and C.T.L. supervised the animal study and provided important experimental insights.

## Competing interests

T.D.N. has a conflict of interest with PiezoBioMembrane Inc. and SingleTimeMicroneedles Inc. Eli Curry is currently an employee and common stock owner of Eli Lilly and Company. The work presented herein was completed prior to his employment with Eli Lilly and Company. Eli Curry is acting entirely on his own and any opinions or endeavors expressed herein are not in any manner affiliated with Eli Lilly and Company. T.V and T.D.N are inventors of the patent application PCT/US2021/053887. The remaining authors declare no competing interests.

## Additional information

[1]Department of Biomedical Engineering, University of Connecticut, Storrs, CT 06269, USA. [2]The Cato T. Laurencin Institute for Regenerative Engineering, University of Connecticut Health, Farmington, CT 06030, USA. [3]Department of Chemical & Biomolecular Engineering, University of Connecticut, Storrs, CT 06269, USA. [4]Department of Mechanical Engineering, University of Connecticut, Storrs, CT 06269, USA. [5]Center of Digital Dentistry/Department of Prosthodontics/Central Laboratory, Peking University School and Hospital of Stomatology & National Center for Stomatology & National Clinical Research Center for Oral Diseases & National Engineering Research Center of Oral Biomaterials and Digital Medical Devices & Beijing Key Laboratory of Digital Stomatology & NHC Research Center of Engineering and Technology for Computerized Dentistry & NMPA Key Laboratory for Dental Materials, Beijing 100081, PR China. [6]Eli Lilly and Company, 450 Kendall Street, Cambridge, MA 02142, USA. [7]Department of Pathobiology and Veterinary Science, University of Connecticut, 61 North Eagleville Road, Unit 3089, Storrs, CT 06269, USA. [8]Pathology and Laboratory Medicine, University of Connecticut Health Center, 63 Farmington Avenue, Farmington, CT 06030, USA. [9]Department of Advanced Manufacturing for Energy Systems Engineering, University of Connecticut, Storrs, CT 06269, USA. [10]Department of Materials Science and Engineering & Institute of Materials Science, University of Connecticut, 25 King Hill Road, Unit 3136, Storrs, CT 06269-3136, USA. [11]Center for Clean Energy Engineering, University of Connecticut, Storrs, CT 06269, USA. [12]Center for Comparative Medicine, University of Connecticut Health Center, Farmington, CT, USA. [13]Institute of Materials Science, University of Connecticut, Storrs, CT 06269, USA. [14]Department of Medicine, University of Connecticut Health Center, Farmington, CT 06030, USA. [15]Department of Orthopaedic Surgery University of Connecticut Health, Farmington, CT 06030, USA. [16]Present address: Department of Biomedical Engineering, University of Connecticut, Storrs, CT 06269, USA. ✉e-mail: nguyentd@uconn.edu

