## [Peer Review File · Nature Communications]

Reviewers' Comments:

Reviewer #1:

Remarks to the Author:

The authors developed an injectable and biodegradable piezoelectric hydrogel with ultrasound activation to offer a minimally invasive regenerative engineering approach for osteoarthritis treatment. The piezoelectric hydrogel can be injected into the joints and self-produce localized electrical cues under ultrasound activation to drive cartilage healing. It is an interesting work, and would offer new insight to the research area. However, there are several major points that should be improved in order to be accepted. Detailed comments are listed as follows:

1. Since the concept of piezoelectric hydrogel based on doping piezoelectric nanomaterials into the hydrogels and the biomedical application of piezoelectric hydrogels have been published in several papers, the title of this work is improper. I suggest to replace it with a more specific one.
2. For the measurement of piezoelectric output under ultrasound activation, the authors only test the ultrasonic response of the nanofiber membrane. What is the exact electrical output (both output voltage and output current) of the composite hydrogel under ultrasound activation?
3. In Figure 2a-d, the expression of four gene shows different trends in the Non-Piezo, Piezo, Non-Piezo+US and Piezo+US groups. Please explain the possible reason in details.
4. As shown on Figure 2g-i, the relative gene expression changes with different concentrations of NF-sPLLA. What is the piezoelectric output of the piezoelectric hydrogels with different concentrations of NF-sPLLA under ultrasound activation?
5. How the authors choose the parameters of ultrasound activation for in vivo treatment. I wonder that how much electrical output or stimulation dose is effective for chondrogenesis is still unknown.
6. What is the safe threshold for electrical stimulation of cartilage repair? And how to determine the safe parameters for piezoelectric stimulation?

Reviewer #2:

Remarks to the Author:

The research paper proposed an injectable piezoelectric hydrogel which could be injected into the joints directly. Then, ultrasound was incorporated to activate the piezoelectric hydrogel, generating electrical stimulation for cartilage regeneration. The utilization of this injectable hydrogel presents a minimally invasive procedure for implantation, decreasing the potential risks triggered by conventional invasive surgery. The strategy used for implantation is novel and the electrical stimulation effects of cartilage regeneration enabled via piezoelectric hydrogel-ultrasound system is satisfactory. Therefore, the manuscript deserves attention for publication. Nevertheless, some concerns listed below need to be handled properly before publication.

1. The title of the manuscript can be re-considered so the main content can be reflected precisely while the novelty is highlighted. For instance, there is no experimental evidence in this manuscript to support the "biodegradable" and the other "medical applications" (except the cartilage regeneration), despite the injectable piezoelectric hydrogel and this implantation strategy certainly have these potentials.
2. Section "Piezoelectric hydrogel for chondrogenesis in vitro study": In this section, three different concentrations of NF-sPLLA in hydrogel, i.e., 1, 5, and 10 mg/ml, were given to explore the optimal concentration for subsequent in vitro and in vivo biomedical experiments. As demonstrated in the manuscript: "At a higher amount of NF-sPLLA, there were more ACAN and SOX9 generated (increasing from 1.5 to 10-fold, from 1 mg/ml to 10 mg/ml). However, for COL2A1 gene, 5 mg/ml of the NF-sPLLA generated more COL2A1 genes compared to other groups.", the 10 mg/mL seems to be more effective compared to 5 mg/mL since the group of 10 mg/mL can induce ACAN and SOX9 genes to the optimal state while only the COL2A1 gene exhibits the optimal state when 5 mg/mL is employed. Nevertheless, as elucidated in the manuscript: "Since collagen II is the most abundant (~ 95%) and important protein in the hyaline cartilage matrix, we selected NF-sPLLA concentration of 5 mg/ml for all following experiments and in vivo studies.", the group of 5 mg/mL was selected for subsequent experiments according to the collagen II. How can this contradiction be explained? Another relevant concern is that I fail to find the data with respect to collagen II in

the section discussing the optimal concentration.

3. Section "Piezoelectric hydrogel induces cartilage healing in rabbit osteochondral defect model":

In the section of animal experiment, three different sensors of polyimide, PLLA, PZT were selected at the beginning to confirm whether the 40 kHz ultrasound could penetrate through tissue and reach the target site. Why don't choose the piezoelectric PLLA and non-piezoelectric PDLLA based hydrogels which have been used to carry out the following animal experiment? The result can also be the direct evidence of that the ultrasound certainly activate the piezoelectric hydrogel to generate electrical signal. When it comes to the data obtained from the ultrasound activated polyimide, PLLA, PZT sensors, as illustrated in Figure S6.b, the difference of output voltage was observed between PLLA and PZT. Nevertheless, it's been reported that PLLA and PZT have a huge disparity on the capability of electromechanical conversion. To be specific, PLLA features a piezoelectric constant of ~ 10 pC/N (<https://doi.org/10.1002/adma.201802084>) while the commercial PZT have a piezoelectric constant of 450–650 pC/N (<https://doi.org/10.1093/nsr/nwac101>). Under the activation of the ultrasound with same intensity and frequency, a more remarkable difference should be obtained on the output voltage of PLLA and PZT. However, the difference of output voltage demonstrated in S6.b is not that large. Can this phenomenon be explained as well?

4. Section "Conclusion/Outlook":

At the end of this section, three different advantages of the research were concluded, as illustrated in the manuscript: "we have presented, for the first time, a novel piezoelectric hydrogel which can (1) be injected into the body via a minimally invasive process to preclude 440 implantation surgery, (2) self-generate electrical cues to promote cartilage and other tissue healing under US activation, and (3) eventually, degrade into safe degradation byproducts to avoid invasive removal surgery and any harm to the body." However, the corresponding data supports over the concluded advantage (3) and part of (2) (i.e., "other tissue healing") are absent, hence, these parts can't be involved in the conclusion as the highlights.

5. The whole manuscript should be carefully rechecked to remove format errors and typos. For example, as illustrated in Figure 1.d, the numbers on the left axis are covered. Furthermore, there is an inconsistency between the scale bars on Figure 1.b and Figure 1.g-i.

Reviewer #3:

Remarks to the Author:

The goal of this study is to evaluate the benefits of an injectable, biodegradable piezoelectric hydrogel for cartilage repair. It is interesting, building up on previous work (ref. 55). However, I have the following points that need critical attention:

1. in vitro:

1a. the authors should have used rabbit BM-MSCs instead of ADSCs as those are the local regenerative cells that will repopulate the defects following gel injection.

1b. I could not find the details on the US conditions employed. Also, do they match with those applied in vivo? How can they be standardized so they would match?

1c. No data are provided on live/dead cells upon (i) US treatment, (ii) hydrogel application, and (iii) both treatments.

2. in vivo:

2a. I could not find the details on the amount of hydrogel applied. Does it match with the condition used in vitro? How can it be standardized so both would match (especially in terms of cell numbers: cell numbers in vitro to BM-derived regenerative cells in vivo)?

2b. Figure 5a: it is hard to understand how the US treatment on itself allows for a better adhesion of the hydrogels to the defects (regardless of piezo/non-piezo or even control); the hydrogel is made of unnatural compounds (PLLA, PDLLA, type-I collagen and not hyaline type-II collagen) so it is again difficult to understand that repair occurs with so much matrix formation just upon US (US-piezo-safO-2 months) versus non-US-piezo-safO-2 months. Non-piezo without US is better over time than with US, please explain. Piezo without US is worse over time, please explain (that would not be good if US do not work or are not well tolerated in patients).

2c. So, please consider the following:

Also here, no data are provided on live/dead cells upon (i) US treatment, (ii) hydrogel application,

and (iii) both treatments (TUNEL assay? Caspase assay?).

Please include an evaluation of the expression of key matrix compounds (collagens II, I, and X for hypertrophy).

Will the US conditions be applicable in patients, will they work, will they be deleterious?

Thanh D. Nguyen, Associate Professor
University of Connecticut
Department of Mechanical Engineering
Room 356 UTEB, Unit 3139 Storrs, CT 06269
nguyentd@uconn.edu · (860) 486-2415

July 27, 2023

Re: Revision for Manuscript, Nature Communications NCOMMS-23-11040-T

Dear Reviewers:

Thank you for your valuable and positive feedback. We have revised our manuscript carefully according to your advice and helpful comments. We have performed additional experiments to obtain the necessary data. We have updated new results in the revised manuscript (both the Main Text and Supplementary information), as marked in red color. Our point-to-point responses are below and the revised manuscript is enclosed.

Reviewer #1:

The authors developed an injectable and biodegradable piezoelectric hydrogel with ultrasound activation to offer a minimally invasive regenerative engineering approach for osteoarthritis treatment. The piezoelectric hydrogel can be injected into the joints and self-produce localized electrical cues under ultrasound activation to drive cartilage healing. It is an interesting work and would offer new insight to the research area. However, there are several major points that should be improved in order to be accepted. Detailed comments are listed as follows:

Response: We sincerely thank the reviewer for understanding the significance of our work and the positive feedback. We greatly appreciate your comments and questions that have helped us to significantly improve our manuscript. We have addressed all comments, as seen below.

Comment #1: Since the concept of piezoelectric hydrogel based on doping piezoelectric nanomaterials into the hydrogels and the biomedical application of piezoelectric hydrogels have been published in several papers, the title of this work is improper. I suggest to replace it with a more specific one.

Response: We thank the reviewer for their comment. We have revised the title to: **“Injectable And Biodegradable Piezoelectric Hydrogel For Osteoarthritis Treatment”**

Comment #2: For the measurement of piezoelectric output under ultrasound activation, the authors only test the ultrasonic response of the nanofiber membrane. What is the exact electrical output (both output voltage and output current) of the composite hydrogel under ultrasound activation?

Response: We thank reviewers for their comment. For clarification, the data presented in the original submission, **Figure 1.e and 1.f** describe the piezoelectric output voltage of the vacuum-dried hydrogel composite. We apologize for the confusion in the legend of **Figure 1.e and f**. We have revised this legend to avoid misinterpretation. We also included the figures here to make it easier for the reviewer to understand.

As seen in **Figure 1.e** the NF-sPLLA dried hydrogel sensor generates a clear signal with consistent intervals and peak magnitude. Meanwhile, NF-sPDLLA dried hydrogel sensor's waveform has smaller amplitude and irregularity with random peaks under the same applied US intensity. **Figure 1.f** indicates that the output voltage of NF-sPLLA dried hydrogel scaffold is around 33.7 mV peak-to-peak, and superior to the negative control NF-sPDLLA dried hydrogel scaffolds (~5mV peak-to-peak).

Figure 1. e, Output voltage waveform of sensors made of our dried NF-sPLLA hydrogel scaffold (Piezo sample) and NF-sPDLLA hydrogel (Non-piezo sample) in collagen under US activation. **f**, Peak-to-Peak output voltage of sensors made of our dried scaffold NF-sPLLA (Piezo sample) and NF-sPDLLA (Non-piezo sample) in collagen under US activation (n=4).

It is noteworthy that we utilized 10X PBS solution and NaOH to crosslink collagen hydrogel, making the liquid composite hydrogel become conductive due to the high concentration of salts. Because of the conductivity properties of the composite hydrogel in its liquid form, it is not feasible to directly measure the piezoelectric output of these hydrogels in their original wet form. Hence, we adopted a vacuum drying method to obtain dried forms of the composite hydrogels, which enabled accurate measurement of the piezoelectric output. This approach was demonstrated in our previous publication, where dried scaffolds were utilized to measure the output voltage of 3D piezoelectric scaffolds [1]. We also updated this information in the Materials and Methods section.

1 Liu, Y. *et al.* Exercise-induced piezoelectric stimulation for cartilage regeneration in rabbits. *Science translational medicine* **14**, eabi7282 (2022).

Comment #3: In Figure 2a-d, the expression of four gene shows different trends in the Non-Piezo, Piezo, Non-Piezo+US and Piezo+US groups. Please explain the possible reason in details.

Response: We thank the reviewer for the comment. To clarify, in our study, we selected *COL2A1*, *ACAN*, *SOX9*, and *GAG* as markers to assess chondrogenesis *in vitro*, because they are known to be crucial for cartilage tissue, both at the gene and protein level. Thus, when evaluating the ability of a biomaterial to promote cartilage formation, it is important to observe an increase in all these genes and proteins. Indeed, our data indicates that the *Piezo + US* group significantly upregulated all *COL2A1*, *ACAN*, and *SOX9* genes, and also led to an increase in *GAG* production compared to both the control group and other sham groups. On the other hand, the other groups, including Non-Piezo with and without US, as well as Piezo without US, exhibited only partial upregulation of individual genes (*SOX9*, *ACAN*, or *COL2A1*), but not all three genes, compared to the control group. These results are consistent with previous research, where the presence of fibers alone or solely

introducing ultrasound (US) stimulation did not enhance chondrogenesis [1,2]. Therefore, we believe that the *Piezo* + *US* combination provides the best conditions for promoting chondrogenesis.

Also, at the current stage, this work is a proof of concept, demonstrating the effectiveness of using piezoelectric hydrogel combined with US activation for cartilage regeneration. So, we focused on the outcome of the experimental group rather than investigating the effect of each factor (e.g., US, NF-sPLLA, NF-sPDLLA) or how each of these effects chondrogenesis. In order to provide a deep understanding of gene trends across different groups, further investigation is required and could be out of scope of this work.

- 1 Liu, Y. *et al.* Exercise-induced piezoelectric stimulation for cartilage regeneration in rabbits. *Science translational medicine* **14**, eabi7282 (2022).
- 2 Yang, S. W. *et al.* Does low-intensity pulsed ultrasound treatment repair articular cartilage injury? A rabbit model study. *BMC Musculoskelet Disord* **15**, 36 (2014). <https://doi.org/10.1186/1471-2474-15-36>

Comment #4: As shown on Figure 2g-i, the relative gene expression changes with different concentrations of NF-sPLLA. What is the piezoelectric output of the piezoelectric hydrogels with different concentrations of NF-sPLLA under ultrasound activation?

Response: Thank you for the reviewer's comment. As suggested, we performed experiments to measure voltage output of different concentrations of NF-sPLLA dried hydrogel sensors under ultrasound (US) activation. The discussion of these data was added to the main text of manuscript and the figure was updated to the **Supplementary Figure 5**.

In these experiments, output voltage of the sensors was measured at 40 KHz and 0.33 Watt/cm² which was the same condition with *in vitro* and *in vivo* studies. However, due to the high electromagnetic interference (EMI) noise at 40 KHz, we further validated the results by measuring the piezoelectric voltage output at 1 MHz, where we were able to control and reduce EMI. **Supplementary Figure 5** depicts that 5mg/ml of NF-sPLLA in hydrogel generated a significantly higher output voltage compared to the 1 mg/ml and 10 mg/ml concentrations. Also, a similar trend of output voltages was observed under 40 KHz. This result indicates that low amounts of NF-sPLLA in the hydrogel produces minimal piezoelectric charges, therefore showing little to no effect on chondrogenesis. On the other hand, an excessive amount of NF-sPLLA within the same volume of hydrogel leads to a high density of fibers. This high fiber density could increase the fiber membrane mass, reducing the mechanical vibration and/or likely cause the charges generated by the fibers to cancel each other out, leading to a reduction in the overall piezoelectric output voltage under US activation. Therefore, 5mg/ml NF-sPLLA in hydrogel is an optimal condition that provides the highest voltage output under US stimulation.

Supplementary Figure 5 | Output voltage V_{pp} of dried NF-sPLLA hydrogel sensors at various concentrations subject to **a**, 40 KHz ($n=3$), the data are expressed as Mean \pm SEM value. $**p < 0.01$, $***p < 0.0001$ and **b**, 1 MHz ($n=3$), the data are expressed as Mean \pm SEM value. $****p < 0.0001$.

Comment #5: How the authors choose the parameters of ultrasound activation for in vivo treatment. I wonder that how much electrical output or stimulation dose is effective for chondrogenesis is still unknown.

Response: Thank you for your comment. We have provided the explanation in the manuscript in **Supplementary Discussions**.

For clarification, the parameters employed for *in vivo* experiments were maintained identical to those utilized for *in vitro* studies, consisting of a 40 KHz ultrasound (US), 0.33 Watt/cm² and exposure for a duration of 20 minutes. These parameters were chosen based on the following reasons:

- First, although 1-3 MHz US frequencies are commonly utilized for US therapy, to penetrate through knee joint and activate the piezoelectric properties of NF-sPLLA hydrogel, a low frequency (e.g., 40 kHz) is more suitable [1]. This is because a lower tissue absorption rate is observed at lower frequencies [2]. Regardless of the frequency employed, it is crucial to ensure that the intensity remains below 0.5 Watt/cm², as low-intensity US which is considered safe for human use [3-6].
- Second, the *in vitro* data (**Figure 2. a-f**) clearly demonstrates that the chosen US parameters were efficient in activating electrical charge in the Piezo hydrogel. This efficiency is evidenced

by the upregulation of gene expressions (*COL2A*, *ACAN*, and *SOX9*), as well as the increased formation of GAG and Collagen II protein in the *Piezo + US* group, when compared with the control/sham groups.

- Third, we also verified that the same US parameters applied in our study effectively activated the piezoelectric charge within the knee joints, as illustrated in **Supplementary Figure 7.b**. This additional evidence further supports the rationale behind our chosen US parameters for *in vivo* experiments.

Regarding the electrical output or stimulation dose for chondrogenesis, these parameters vary across different studies [7,8]. Currently, there is no clear value on the optimal or effective electrical cue dose for promoting cartilage healing. However, based on our *in vitro* study, we have found that our chosen US parameters and the charge generated from our piezoelectric hydrogel are safe to promote adipose-derived stem cells (ADSCs) proliferation and effective to facilitate their differentiation into chondrocyte cells. Therefore, we have decided to use the same parameters for our *in vivo* study. Nevertheless, this study is currently in the exploratory stage, aiming to establish the proof of concept for our work. Consequently, determining the threshold or optimal dosage for the electrical output falls beyond the scope of this present investigation and requires further investigations.

1. Lucas, V. S., Burk, R. S., Creehan, S. & Grap, M. J. Utility of high-frequency ultrasound: moving beyond the surface to detect changes in skin integrity. *Plast Surg Nurs* **34**, 34-38 (2014). <https://doi.org/10.1097/psn.0000000000000031>
2. Luo L, Molnar J, Ding H, Lv X, Spengler G: **Ultrasound absorption and entropy production in biological tissue: a novel approach to anticancer therapy**. *Diagn Pathol* 2006, **1**:35
3. er Haar, G. Therapeutic ultrasound. *European Journal of Ultrasound* **9**, 3-9 (1999). [https://doi.org/https://doi.org/10.1016/S0929-8266\(99\)00013-0](https://doi.org/https://doi.org/10.1016/S0929-8266(99)00013-0)
4. Khanna, A., Nelmes, R. T. C., Gougoulas, N., Maffulli, N. & Gray, J. The effects of LIPUS on soft-tissue healing: a review of literature. *British Medical Bulletin* **89**, 169-182 (2008). <https://doi.org/10.1093/bmb/ldn040>
5. Baek, H., Pahk, K. J. & Kim, H. A review of low-intensity focused ultrasound for neuromodulation. *Biomedical Engineering Letters* **7**, 135-142 (2017). <https://doi.org/10.1007/s13534-016-0007-y>
6. Xin, Z., Lin, G., Lei, H., Lue, T. F. & Guo, Y. Clinical applications of low-intensity pulsed ultrasound and its potential role in urology. *Transl Androl Urol* **5**, 255-266 (2016). <https://doi.org/10.21037/tau.2016.02.04>
7. Vaca-González, J. J. *et al.* Biophysical Stimuli: A Review of Electrical and Mechanical Stimulation in Hyaline Cartilage. *Cartilage* **10**, 157-172 (2019). <https://doi.org/10.1177/1947603517730637>
8. Baker, B., Spadaro, J., Marino, A. & Becker, R. O. Electrical stimulation of articular cartilage regeneration. *Ann N Y Acad Sci* **238**, 491-499 (1974). <https://doi.org/10.1111/j.1749-6632.1974.tb26815.x>

Comment #6: What is the safe threshold for electrical stimulation of cartilage repair? And how to determine the safe parameters for piezoelectric stimulation?

Response: We thank the reviewer for their comment.

Electrical stimulation (ES) could adversely affect tissue in several ways, such as electrical burns, irreversible electroporation, and electric shock [1]. Consequently, caution must be exercised when utilizing ES for tissue engineering, as excessive ES can harm the body. The configurations of ES involve various factors, including field strength, stimulation duration, and the type of ES, such as direct current (DC) (directly contacted DC and capacitive coupling) or biphasic current (pulses and alternating current). Therefore, defining a safe threshold depends on the specific configuration of ES

applied. However, there is currently a lack of clear guidelines or comprehensive studies identifying/evaluating safe threshold parameters for ES use in tissue regeneration, particularly in cartilage healing.

To establish safe parameters for piezoelectric stimulation, careful consideration of various factors is required. Firstly, for piezoelectricity activation methods, vibration intensity or mechanical pressure applied to piezoelectric material should fall within a range that mitigates the risk of cartilage damage. Secondly, the magnitude of voltage output generated by the piezoelectric stimulation must adhere to the safe range for ES. In this regard, for US intensity, we utilized low intensity (0.33 watt/cm^2) which is safe for human use [2-5]. Furthermore, with the intensity of US employed in our study, the voltage output generated is very low and comparable to that observed in Barker's study, which utilized ES (ranging from 15 to 500 mV) for cartilage regeneration in a rabbit model [6]. On top of that, our *in vitro* data indicates that the US intensity and the resulting output voltage applied to ADSCs are biocompatible (**Supplementary Figure 3.c**) and do not cause any harm to rabbits after a two-month treatment period (**Supplementary Movie 4**). Collectively, we believe the parameter for piezoelectric stimulation applied in this study is safe.

The manuscript was revised that included this information.

1. Bikson, M. A review of hazards associated with exposure to low voltages. *New York: University of New York* **20** (2004).
2. ter Haar, G. Therapeutic ultrasound. *European Journal of Ultrasound* **9**, 3-9 (1999). [https://doi.org/10.1016/S0929-8266\(99\)00013-0](https://doi.org/10.1016/S0929-8266(99)00013-0)
3. Khanna, A., Nelmes, R. T. C., Gougoulias, N., Maffulli, N. & Gray, J. The effects of LIPUS on soft-tissue healing: a review of literature. *British Medical Bulletin* **89**, 169-182 (2008). <https://doi.org/10.1093/bmb/ldn040>
4. Baek, H., Pahk, K. J. & Kim, H. A review of low-intensity focused ultrasound for neuromodulation. *Biomedical Engineering Letters* **7**, 135-142 (2017). <https://doi.org/10.1007/s13534-016-0007-y>
5. Xin, Z., Lin, G., Lei, H., Lue, T. F. & Guo, Y. Clinical applications of low-intensity pulsed ultrasound and its potential role in urology. *Transl Androl Urol* **5**, 255-266 (2016). <https://doi.org/10.21037/tau.2016.02.04>
6. Baker, B., Spadaro, J., Marino, A. & Becker, R. O. Electrical stimulation of articular cartilage regeneration. *Ann N Y Acad Sci* **238**, 491-499 (1974). <https://doi.org/10.1111/j.1749-6632.1974.tb26815.x>

Reviewer #2:

Comments: The research paper proposed an injectable piezoelectric hydrogel which could be injected into the joints directly. Then, ultrasound was incorporated to activate the piezoelectric hydrogel, generating electrical stimulation for cartilage regeneration. The utilization of this injectable hydrogel presents a minimally invasive procedure for implantation, decreasing the potential risks triggered by conventional invasive surgery. The strategy used for implantation is novel and the electrical stimulation effects of cartilage regeneration enabled via piezoelectric hydrogel-ultrasound system is satisfactory. Therefore, the manuscript deserves attention for publication. Nevertheless, some concerns listed below need to be handled properly before publication.

Response: We sincerely thank the reviewer for reading our manuscript carefully and the highly positive remarks. Furthermore, we appreciate the reviewer's suggestions, which helped us improve our manuscript. Please find below our detailed responses to the reviewer.

Comment #1: The title of the manuscript can be re-considered so the main content can be reflected precisely while the novelty is highlighted. For instance, there is no experimental evidence in this manuscript to support the “biodegradable” and the other “medical applications” (except the cartilage regeneration), despite the injectable piezoelectric hydrogel and this implantation strategy certainly have these potentials.

Response: We thank the reviewer for their feedback. Upon careful reconsideration of the manuscript's title, we changed to a new title “**Injectable And Biodegradable Piezoelectric Hydrogel For Osteoarthritis Treatment**”

We have retained the key term "biodegradable" in the title, as our hydrogel consists of collagen I and PLLA, both widely recognized as biodegradable materials [1,2]. Additionally, our previous research demonstrated that scaffolds fabricated from collagen I and PLLA fibers degrade over time [3]. In the revised version of the manuscript, we have also performed the degradation study of the NF-sPLLA hydrogel taken at 37°C over a period of 9 weeks, along with an accelerated degradation condition (80°C), as supporting evidence of its degradability. As seen in **Supplementary Figure 2.g**, the volume of the NF-sPLLA hydrogel scaffolds gradually decreased over time. After week 9, under accelerated conditions, the hydrogels degraded, broke down, and lost their original structures.

Supplementary Figure 2. g, Degradation study of NF-sPLLA hydrogel at 37 °C and accelerated condition 80 °C both in media and PBS with and without US treatment (scale bar: 1 cm).

1. Okada, T., Hayashi, T. & Ikada, Y. Degradation of collagen suture in vitro and in vivo. *Biomaterials* **13**, 448-454 (1992).
2. Le, T. T. *et al.* Piezoelectric nanofiber membrane for reusable, stable, and highly functional face mask filter with long-term biodegradability. *Advanced Functional Materials* **32**, 2113040 (2022).
3. Liu, Y. *et al.* Exercise-induced piezoelectric stimulation for cartilage regeneration in rabbits. *Science translational medicine* **14**, eabi7282 (2022).

Comment #2: Section “Piezoelectric hydrogel for chondrogenesis in vitro study”: In this section, three different concentrations of NF-sPLLA in hydrogel, i.e., 1, 5, and 10 mg/ml, were given to explore the optimal concentration for subsequent in vitro and in vivo biomedical experiments. As demonstrated in the manuscript: “At a higher amount of NF-sPLLA, there were more ACAN and SOX9 generated (increasing from 1.5 to 10-fold, from 1 mg/ml to 10 mg/ml). However, for COL2A1 gene, 5 mg/ml of the NF-sPLLA generated more COL2A1 genes compared to other groups.” The 10 mg/mL seems to be more effective compared to 5 mg/mL since the group of 10 mg/mL can induce ACAN and SOX9 genes to the optimal state while only the COL2A1 gene exhibits the optimal state when 5 mg/mL is employed. Nevertheless, as elucidated in the manuscript: “Since collagen II is the most abundant (~ 95%) and important protein in the hyaline cartilage matrix, we selected NF-sPLLA concentration of 5 mg/ml for all following experiments and in vivo studies.”, the group of 5 mg/mL was selected for subsequent experiments according to the collagen II. How can this contradiction be explained?

Another relevant concern is that I fail to find the data with respect to collagen II in the section discussing the optimal concentration.

Response: We thank the reviewer for the comment and sorry for the confusion. For clarification, we selected 5mg/ml concentration for subsequent experiments because of the highest production of collagen II and the overall best piezoelectric performance of this scaffold.

Collagen II is the main component of cartilage, which constitutes up to 95% of the collagens in the cartilage. Studies have revealed that *COL2A1* functions as an extracellular signaling molecule capable of significantly suppressing chondrocyte hypertrophy by promoting integrin $\beta 1$ –SMAD1 interaction [1-3], which avoids cartilage calcification. This is important because in the process of regenerating articular cartilage, it is not only necessary for cells to differentiate into chondrocytes but also for them to stably maintain the hyaline cartilage stage, which is distinct from the growth plate zone. Additionally, *COL2A1* is considered as an important extracellular signaling molecule that can regulate chondrocyte proliferation and metabolism, similar to soluble molecule signals [4,5]. Moreover, the collagen II network plays a vital role in retaining proteoglycans within the cartilage matrix and is the most essential protein in the hyaline cartilage matrix [6].

In addition, per reviewer 1 suggestion, we assessed the piezoelectric performance of the hydrogel at different concentrations under ultrasound (US) stimulation. In these experiments, NF-sPLLA dried hydrogel sensors at various concentrations output voltage were measured at 40 KHz and 0.33 Watt/cm² which was the same condition with *in vitro* and *in vivo* studies. However, due to the high electromagnetic interference noise at 40 KHz, we further validated the results by measuring the piezoelectric voltage output at 1 MHz. **Supplementary Figure 5** depicts that 5mg/ml of NF-sPLLA in hydrogel generated a significantly higher output voltage compared to the other 1 mg/ml and 10 mg/ml under both 40 KHz and 1 MHz.

Supplementary Figure 5| Output voltage Vpp of dried NF-sPLLA hydrogel sensors at various concentrations subject to a, 40 KHz (n=3), the data are expressed as Mean \pm SEM value. **p < 0.01, *p < 0.0001 and b, 1 MHz (n=3), the data are expressed as Mean \pm SEM value. ****p < 0.0001.**

Collectively, 5mg/ml of NF-sPLLA hydrogel which provided the best overall piezo performance (compared to higher concentration 10 mg/ml nanofiber hydrogel), and produced the most COL2A1 gene expression was selected for all subsequent experiments and *in vivo* studies.

We already edited the main text to add the information.

1. Freyria, A.-M. & Mallein-Gerin, F. Chondrocytes or adult stem cells for cartilage repair: the indisputable role of growth factors. *Injury* **43**, 259-265 (2012).
2. Cuervo, B. *et al.* Hip osteoarthritis in dogs: a randomized study using mesenchymal stem cells from adipose tissue and plasma rich in growth factors. *International journal of molecular sciences* **15**, 13437-13460 (2014).
3. Lian, C. *et al.* Collagen type II suppresses articular chondrocyte hypertrophy and osteoarthritis progression by promoting integrin β 1– SMAD1 interaction. *Bone research* **7**, 8 (2019)
4. Xin, W., Heilig, J., Paulsson, M. & Zaucke, F. Collagen II regulates chondrocyte integrin expression profile and differentiation. *Connective Tissue Research* **56**, 307-314 (2015).
5. Klatt, A. R. *et al.* Discoidin domain receptor 2 mediates the collagen II-dependent release of interleukin-6 in primary human chondrocytes. *The Journal of Pathology: A Journal of the Pathological Society of Great Britain and Ireland* **218**, 241-247 (2009).
6. Lavietes, B. B. Kinetics of matrix synthesis in cartilage cell cultures. *Experimental cell research* **68**, 43-48 (1971).

Comment #3: Section “Piezoelectric hydrogel induces cartilage healing in rabbit osteochondral defect model”: In the section of animal experiment, three different sensors of polyimide, PLLA, PZT were selected at the beginning to confirm whether the 40 kHz ultrasound could penetrate through tissue and reach the target site. Why don’t choose the piezoelectric PLLA and non-piezoelectric PDLA based hydrogels which have been used to carry out the following animal experiment? The result can also be the direct evidence that the ultrasound certainly activate the piezoelectric hydrogel to generate

electrical signal. When it comes to the data obtained from the ultrasound activated polyimide, PLLA, PZT sensors, as illustrated in Figure S6.b, the difference of output voltage was observed between PLLA and PZT. Nevertheless, it's been reported that PLLA and PZT have a huge disparity on the capability of electromechanical conversion. To be specific, PLLA features a piezoelectric constant of ~ 10 pC/N (<https://doi.org/10.1002/adma.201802084>) while the commercial PZT have a piezoelectric constant of 450–650 pC/N (<https://doi.org/10.1093/nsr/nwac101>). Under the activation of the ultrasound with same intensity and frequency, a more remarkable difference should be obtained on the output voltage of PLLA and PZT. However, the difference of output voltage demonstrated in S6.b is not that large. Can this phenomenon be explained as well?

Response: We thank the reviewer for their comment. We would like to clarify that **the purpose of this experiment is only to validate the penetration ability of 40 kHz US through various tissues, including skin, muscle, and ligament, and reach the targeted defect site.** We selected lead zirconate titanate (PZT) as a positive control because PZT is a commonly used piezoelectric material for many medical applications. It is important to note that when using a 40 KHz US transducer to measure piezoelectric response of materials, the carried-out data may tangle with electromagnetic interference (EMI) noise. Therefore, we utilized non-piezoelectric material (e.g., polyimide) to generate a baseline which is only subjective to EMI but does not exhibit piezoelectric properties. As seen in **Supplementary Figure 7.b** the polyimide sensor also shows some signals at 40 kHz, but purely EMI noise and not piezoelectric signal. However, PLLA sensor (made of the aligned nanofiber film) which has comparable dielectric constant to polyimide (2.7 for PLLA and ~ 3 for polyimide), produced significantly higher output voltage due to their piezoelectricity properties. Regardless, **Supplementary Figure 7.b** demonstrates that the 40 kHz US can effectively penetrate different tissues and reach the intended defect site, successfully activating the piezoelectric response of the PZT, PLLA sensors.

We did not use NF-sPLLA and NF-sPDLLA-based hydrogel sensors to carry out the experiments for the following reasons. (1) We were using a 40 KHz US transducer, and there is considerable amount of EMI noise that would interfere with the measured signals which are small from the dried hydrogels. (2) The fibers inside the sensors made from the dried NF-sPLLA or NF-sPDLLA based hydrogels (i.e., chopped nanofibers mixed inside the collagen and dried) were separated from each other and randomly oriented; therefore, these sensors have much less piezoelectricity (in the bulk material) compared to the sensors fabricated from the aligned PLLA nanofiber films (i.e. the non-chopped nanofiber film as it is after the electrospinning process). Besides, the sensors used in these experiments were significantly small in size (5x5mm) to ensure they fit inside the knee joint defect. This small size further reduced the output signals. Therefore, the dried NF-sPLLA and NF-sPDLLA-based hydrogel sensors are not ideal for confirming 40 KHz US penetration as they both would produce very small response signals due to the low piezoelectric effect, the EM noise, and the small device size. It should be noted that we already provided evidence of the piezoelectric properties of NF-sPLLA hydrogel and the non-piezoelectric properties of NF-sPDLLA in **Figure 1.e and f** (in the original submission).

In **Supplementary Figure 7.b**, the data with PZT, PLLA and polyimide already served our purpose of verifying that the selected US frequency and intensity can penetrate the rabbit's knee joint and reach the defect. Hence, we believe it is unnecessary to repeat the experiment with the dried NF-sPLLA and NF-sPDLLA based hydrogel sensors.

Regarding the small difference between PZT and PLLA sensors, we agree that PZT possesses greater piezoelectric constants than ones of PLLA. This often leads to an assumption that PZT always produces a much higher output than PLLA. However, this is only true when the materials are stimulated by impact forces at low frequencies. For ultrasound transmission, especially for responding to the US, the output performance of the piezoelectric materials depends strongly on their acoustic impedance, which defines how well the US can be transmitted between different mediums (in our case, between tissues and the testing sensor). Indeed, PZT acoustic impedance is very high (34.7 MRayl) compared to the averaged acoustic impedance of tissues (~1.5 MRayl), leading to a major amount of US scattered or reflected to surrounding tissues instead of stimulating the material. In fact, for practical applications, PZT-based ultrasound transducers require a matching layer and a backing layer to receive/respond to the US effectively. On the other hand, PLLA's acoustic impedance (~2.3 MRayl) is closer to one of the body tissues, allowing more US to activate the materials. Therefore, the signal from the PLLA sensors is slightly smaller than PZT sensors in our ultrasound measurement.

This explanation was included in the **Supplementary Discussions**.

Comment #4: Section “Conclusion/Outlook”: At the end of this section, three different advantages of the research were concluded, as illustrated in the manuscript: “we have presented, for the first time, a novel piezoelectric hydrogel which can (1) be injected into the body via a minimally invasive process to preclude implantation surgery, (2) self-generate electrical cues to promote cartilage and other tissue healing under US activation, and (3) eventually, degrade into safe degradation byproducts to avoid invasive removal surgery and any harm to the body.” However, the corresponding data supports over the concluded advantage (3) and part of (2) (i.e., “other tissue healing”) are absent, hence, these parts can't be involved in the conclusion as the highlights.

Response: We thank the reviewer for their comment. We have edited our writing to make the precise conclusion as presented below:

“we have presented, for the first time, a novel piezoelectric hydrogel which can (1) be injected into the body via a minimally invasive process to preclude implantation surgery, (2) self-generate electrical cues to promote cartilage and **potentially heal** other tissues under US activation, and (3) eventually, degrade into safe degradation byproducts to avoid invasive removal surgery and any harm to the body.”

We kept claim (3) as this is explained and data is provided in comment #1.

Comment #5: The whole manuscript should be carefully rechecked to remove format errors and typos. For example, as illustrated in Figure 1.d, the numbers on the left axis are covered. Furthermore, there is an inconsistency between the scale bars on Figure 1.b and Figure 1.g-i.

Response: We thank the reviewer for their comment. We have revised all the figures and carefully revised the manuscript to remove errors and typos.

Reviewer #3:

Comments: The goal of this study is to evaluate the benefits of an injectable, biodegradable piezoelectric hydrogel for cartilage repair. It is interesting, building up on previous work (ref. 55). However, I have the following points that need critical attention:

Response: We thank the reviewer for supportive feedback. In addition, we greatly appreciate your comments and questions that have helped us to significantly improve our manuscript. We have addressed all comments, as seen below.

***In vitro* comments:**

Comment #1: The authors should have used rabbit BM-MSCs instead of ADSCs as those are the local regenerative cells that will repopulate the defects following gel injection.

Response: We thank the reviewer for the comment. We agree that BM-MSCs are the local regenerative cells that likely will repopulate the cartilage defects following gel injection. However, scientific evidence has demonstrated that both adipose-derived stem cells (ADSCs) and BM-MSCs possess equivalent potential for differentiation into various tissue lineages, including cartilage, bone, and skeletal muscle [1-3]. Furthermore, ADSCs offer distinct advantages, including their availability, accessibility and the ease to be expanded for *in vitro* experiments [4]. On top of that, we want to emphasize that ADSCs were chosen just as a stem cell model to validate our hypothesis that piezoelectric charge can induce stem cells (in general) into chondrocyte phenotype *in vitro* (in comparison with the other control groups of using non-piezoelectric stimulation). Therefore, as long as we use the same stem cell source and the same cell-culture condition for all *in vitro* groups, our experiment outcomes still serve our purpose to indicate the effect of piezoelectric stimulation on chondrogenesis.

We also revised our manuscript with this clear explanation provided here.

1. Schäffler, A. & Büchler, C. Concise review: adipose tissue-derived stromal cells—basic and clinical implications for novel cell-based therapies. *Stem cells* **25**, 818-827 (2007).
2. Wei, Z. *et al.* Bone marrow mesenchymal stem cells from leukemia patients inhibit growth and apoptosis in serum-deprived K562 cells. *Journal of Experimental & Clinical Cancer Research* **28**, 1-7 (2009).
3. Izadpanah, R. *et al.* Biologic properties of mesenchymal stem cells derived from bone marrow and adipose tissue. *Journal of cellular biochemistry* **99**, 1285-1297 (2006).
4. Simonacci, F., Bertozzi, N., Grieco, M. P. & Raposio, E. From liposuction to adipose-derived stem cells: indications and technique. *Acta Biomed* **90**, 197-208 (2019). <https://doi.org/10.23750/abm.v90i2.6619>
5. Qin, Y. *et al.* An Update on Adipose-Derived Stem Cells for Regenerative Medicine: Where Challenge Meets Opportunity. *Advanced Science* **n/a**, 2207334 <https://doi.org/https://doi.org/10.1002/advs.202207334>
6. Lee, S. *et al.* ADSC-Based Cell Therapies for Musculoskeletal Disorders: A Review of Recent Clinical Trials. *Int J Mol Sci* **22** (2021). <https://doi.org/10.3390/ijms221910586>

Comment #2: I could not find the details on the US conditions employed. Also, do they match with those applied *in vivo*? How can they be standardized so they would match?

Response: We thank the reviewer for the comment. For clarification, this information was provided in the Material and Methods of the original manuscript.

To maintain consistency between the *in vivo* and *in vitro* experiments, we applied the same parameters that were used *in vitro* to *in vivo*. In details, for *in vitro* treatment, we utilized an ultrasonic bath (Branson 2800 CPX series) that generated US at a frequency of 40 KHz and an intensity of 0.33 Watt/cm². The cells were exposed to this US for a duration of 20 minutes. To replicate these conditions in the *in vivo* study, we developed a similar system consisting of a 40 kHz ultrasound generator (Steminc) equipped with two 40 kHz bolt clamped Langevin transducers (Steminc) connected in series. The same intensity

of US and treatment time were applied in the *in vivo* experiments to make sure comparability with the *in vitro* setup.

Comment #3: No data are provided on live/dead cells upon (i) US treatment, (ii) hydrogel application, and (iii) both treatments.

Response: We thank the reviewer for the comment. This information was provided in **Supplementary Figure 3.c** in the original submission.

Our experimental data reveals that both the Piezo and Non-Piezo groups, with or without US treatment, exhibit biocompatibility comparable to the control group (cells inside collagen only) in both short-term (1-3 days) and long-term (14 days) assessments. Notably, the viability of ADSCs in the Piezo + US group showed a significant increase on day 9 and day 14 compared to the other control/sham groups. This result is consistent with literature indicating that piezoelectric charges/electrical stimulation (ES) had a positive influence on cell growth [1-3]. Additionally, we also performed a hemolysis study, as shown in **Supplementary Figure 3.a-b**, which indicated these hydrogels are highly safe with a low hemolysis rate (less than 5%) to satisfy the requirements of the International Standards Organization.

We also want to emphasize that in this work, we employed low-frequency and low-intensity US, both *in vitro* and *in vivo* study. The parameters of US being employed are safe and optimal knee joint

treatment which has been established as safe and capable of penetrating deep tissues within the knee joint [4,5]. We applied an intensity of 0.33 Watt/cm² which is below 0.5 Watt/cm², as low-intensity US which is deemed safe for human use [6-9].

1. Wang, A., Hu, M., Zhou, L. & Qiang, X. Self-Powered Well-Aligned P(VDF-TrFE) Piezoelectric Nanofiber Nanogenerator for Modulating an Exact Electrical Stimulation and Enhancing the Proliferation of Preosteoblasts. *Nanomaterials (Basel)* **9** (2019). <https://doi.org/10.3390/nano9030349>
2. Chen, C., Bai, X., Ding, Y. & Lee, I.-S. Electrical stimulation as a novel tool for regulating cell behavior in tissue engineering. *Biomaterials Research* **23**, 25 (2019). <https://doi.org/10.1186/s40824-019-0176-8>
3. Xia, G., Wang, G., Yang, H., Wang, W. & Fang, J. Piezoelectric charge induced hydrophilic poly(L-lactic acid) nanofiber for electro-topographical stimulation enabling stem cell differentiation and expansion. *Nano Energy* **102**, 107690 (2022). <https://doi.org/10.1016/j.nanoen.2022.107690>
4. Chan, V. & Perlas, A. in *Atlas of Ultrasound-Guided Procedures in Interventional Pain Management* (ed Samer N. Narouze) 13-19 (Springer New York, 2011).
5. Lucas, V. S., Burk, R. S., Creehan, S. & Grap, M. J. Utility of high-frequency ultrasound: moving beyond the surface to detect changes in skin integrity. *Plast Surg Nurs* **34**, 34-38 (2014). <https://doi.org/10.1097/psn.0000000000000031>
6. ter Haar, G. Therapeutic ultrasound. *European Journal of Ultrasound* **9**, 3-9 (1999). [https://doi.org/10.1016/S0929-8266\(99\)00013-0](https://doi.org/10.1016/S0929-8266(99)00013-0)
7. Khanna, A., Nelmes, R. T. C., Gougoulas, N., Maffulli, N. & Gray, J. The effects of LIPUS on soft-tissue healing: a review of literature. *British Medical Bulletin* **89**, 169-182 (2008). <https://doi.org/10.1093/bmb/ldn040>
8. Baek, H., Pahk, K. J. & Kim, H. A review of low-intensity focused ultrasound for neuromodulation. *Biomedical Engineering Letters* **7**, 135-142 (2017). <https://doi.org/10.1007/s13534-016-0007-y>
9. Xin, Z., Lin, G., Lei, H., Lue, T. F. & Guo, Y. Clinical applications of low-intensity pulsed ultrasound and its potential role in urology. *Transl Androl Urol* **5**, 255-266 (2016). <https://doi.org/10.21037/tau.2016.02.04>

***In vivo* comments:**

Comment #4: I could not find the details on the amount of hydrogel applied. Does it match with the condition used *in vitro*? How can it be standardized so both would match (especially in terms of cell numbers: cell numbers *in vitro* to BM-derived regenerative cells *in vivo*)?

Response: We thank the reviewer for the comment.

In terms of the amount of hydrogel applied *in vivo*, we utilized approximately 30 µl of hydrogel, completely filling the defect, which had a diameter of 4 mm and a depth of 2 mm (considered as a critical-size osteochondral defect in the rabbit's knee).

Regarding of standardizing *in vitro* and *in vivo* condition:

- For cell density and cell number we would like to emphasize that: 1) in this work, we use piezoelectric hydrogel which is cell-free and will generate charges under ultrasound stimulation to promote body's own cells to migrate and heal the defects. Therefore, matching the cells used *in vitro* with the BM-MSCs in the *in vivo* condition is irrelevant. 2) Cell density or cell numbers that was used *in vitro* testing had allowed us to optimize fiber concentrations, confirm the US parameters that yielded positive outcomes, and validate our hypothesis before animal experiments.
- For ultrasound intensity and frequency: we applied the same parameters for both *in vitro* and the *in vivo* experiments, which were 0.33 watt/cm² and 40 KHz, with a treatment time of 20 minutes.

We updated the manuscript which included relevant information.

Comment #5: Figure 5a:

- a. it is hard to understand how the US treatment on itself allows for a better adhesion of the hydrogels to the defects (regardless of piezo/non-piezo or even control);
- b. the hydrogel is made of unnatural compounds (PLLA, PDLA, type-I collagen and not hyaline type-II collagen) so it is again difficult to understand that repair occurs with so much matrix formation just upon US (US-piezo-safO-2 months) versus non-US-piezo-safO-2 months.
- c. Non-piezo without US is better over time than with US, please explain. Piezo without US is worse over time, please explain (that would not be good if US do not work or are not well tolerated in patients).

Response: We thank the reviewer for the comment.

- a. To clarify, the images provided in the manuscript depict H&E and Safran-O staining, which were used to visualize tissue and cellular structures within the defects after 1 or 2 months of treatments. These images do not visualize hydrogels. Furthermore, **our purpose of utilizing US treatment was to remotely activate the piezoelectric charge from NF-sPPLA hydrogel, rather than promoting adhesion of the hydrogel with native tissues. Therefore, US treatment did not have any relation with adhesion of hydrogels and native tissues.** Please note that the matrix formation observed in the control group (defect only) or controls + US is solely the result of the body's attempt to repair the injured cartilage by depositing fibrous scar tissue [1,2].
- b. The use of synthetic compounds as biomaterials for cartilage regeneration has shown promising results in previous research, including alginate, Heparin, and PEG-based hydrogel [1]. Therefore, it is normal to deploy these compounds as scaffolds in tissue engineering. In this study, these compounds (PLLA + type-I collagen) and US were used as the vehicle to deliver electrical cue. Our primary hypothesis is that electrical charges generated by the proposed hydrogel stimulate stem cells differentiation into the chondrocyte phenotype. As shown in **Figure 3**, we believe that the generated piezoelectric charges from the hydrogel also stimulate stem cells to generate TGF- β 1 which is one of important growth factors in cartilage healing. Therefore, so much more hyaline cartilage matrix was formed and integrated well with the native tissues in the *Piezo + US* group (compared to the *Piezo - US* and other groups), which validated our hypothesis. This finding emphasizes the potential of the piezoelectric hydrogel as a promising approach for facilitating cartilage regeneration and repair.
- c. We want to emphasize that the images presented in **Figure 5.a** only show the histological staining (H&E and Safranin-O) of one animal per group, serving as representative data. Since some images could lead to incorrect conclusions at the first glance, we decided to replace those images and revise **Figure 5.a** We would like to elaborate on the representative images of Non-Piezo without US after 1 and 2 months. Even though 2 months had more tissue filling in the defect compared with 1 month, the defect was infiltrated with fibrosis scar tissue (hot pink arrow). Additionally, the fibrosis tissue was detached from native tissue which was similar for 1 month data (violet markers). For the Piezo group, at 1 month time point, we observed

chondrocyte-like cells were formed inside the defect, however, the newly formed tissue was detached from the native tissue and the cells did not pack in any cartilage structure. Therefore, with time, the tissue will either collapse and fall out of defect due to the unstable structure or turn into bony tissue which was similar to 2 months data. Hence, Non-Piezo and Piezo samples without US at 1- or 2-month time point are almost the same in terms of cartilage regeneration. Along with the representative data, **Figure 5.b** (which was provided in the original submission) depicts quantitative histology score for all animals in the groups, evaluated by professional pathologists. Based on the histology scoring data, there were no significant differences in terms of cartilage regeneration between the control and sham groups at both the 1-month and 2-month time points. This suggests that neither US treatment nor the piezoelectric hydrogel alone exerted any notable positive or negative effects on the cartilage healing process in the animals. However, the combination of *Piezo* + *US*, which generated electrical stimulation, demonstrated a significant improvement in the healing process.

Figure 5 | Piezoelectric hydrogel induces cartilage healing, evaluated by histology assessment and mechanical testing *in vivo*. **a**, H&E staining and Safranin O/fast green and collagen II staining to evaluate the articular cartilage regeneration for sham (defect only), non-piezo/piezo hydrogels with and without US activation (1-2 months). Black arrows indicate newly formed cartilage tissues. Yellow markers indicate the new cartilage tissue which was well-integrated with the native host tissue. Hot pink arrows indicate fibrillation filling, red arrows indicate bony tissue and violet markers indicate the detachment of newly formed tissue from the host (scale bars: 500 μ m). **b**, ICRS histological evaluation, (n=4 knees for each group). The score was an average point from three independent professionals and a blinded evaluation. The data are expressed as data points with Mean \pm SEM *p < 0.05.

We revised our manuscript to make our points clearer where it was necessary.

1. Silver, F. H. & Glasgold, A. I. Cartilage Wound Healing: An Overview. *Otolaryngologic Clinics of North America* **28**, 847-864 (1995). [https://doi.org/10.1016/S0030-6665\(20\)30463-1](https://doi.org/10.1016/S0030-6665(20)30463-1)
2. Steinwachs, M. R., Waibl, B. & Mumme, M. Arthroscopic Treatment of Cartilage Lesions With Microfracture and BST-CarGel. *Arthroscopy Techniques* **3**, e399-e402 (2014). <https://doi.org/10.1016/j.eats.2014.02.011>
3. Liu, M. *et al.* Injectable hydrogels for cartilage and bone tissue engineering. *Bone Research* **5**, 17014 (2017). <https://doi.org/10.1038/boneres.2017.14>

Comment #6: So, please consider the following:

- a. Also here, no data are provided on live/dead cells upon (i) US treatment, (ii) hydrogel application, and (iii) both treatments (TUNEL assay? Caspase assay?).
- b. Please include an evaluation of the expression of key matrix compounds (collagens II, I, and X for hypertrophy).
- c. Will the US conditions be applicable in patients, will they work, will they be deleterious?

Response: We thank the reviewer for the comment. We have performed necessary experiments to obtain more data as you suggested.

- a. While the TUNEL assay or caspase assay is utilized to assess cell apoptosis through DNA fragmentation or caspase detection, H&E staining can also provide valuable information about apoptosis based on cell morphology [1-3]. In the early stages of apoptosis, cells undergo a reduction in size as both the cytoplasm and nucleus condense. Subsequently, the nucleus begins to fragment. Additionally, the irreversible nuclear condensation and tightly packed phenomenon are referred to as pyknosis, and the fragmented nucleus is called karyorrhexis [1-5]. These features can be visualized using H&E staining under a light microscope. During the H&E examination, apoptotic cells are identifiable as a mass of dark eosinophilic cytoplasm, adopting round or oval shapes, with tightly packed purple or fragmented nuclear chromatin [1-3]. As shown in **Supplementary Figure 9** (H&E staining at higher magnification), we did not observe any pyknosis or karyorrhexis in any of the groups which indicates that hydrogels and US did not induce cells apoptosis.

Supplementary Figure 9 | Piezoelectric hydrogel induces cartilage healing, evaluated by cell apoptosis using H&E. H&E staining for apoptosis cell visualization at high magnification, for sham (defect only), non-piezo/piezo hydrogels with and without US activation (1-2 months). The apoptotic cells (pyknosis, karyorrhexis) are identifiable as a mass of dark eosinophilic cytoplasm, adopting

round or oval shapes, with tightly packed purple or fragmented nuclear chromatin. The pyknosis or karyorrhexis were not observed in any of groups which indicates that hydrogels and US did not induce cells apoptosis. scale bar: 10 μm .

- b. We have conducted Collagen II staining experiments, and the results are presented in **Figure 5a**. (or Figure below). The data indicates that the experimental groups (*Piezo* + *US*) exhibited highly positive collagen II staining, indicating abundant collagen II protein within the defects. In contrast, the other groups showed minimal to no observable collagen II production. Interestingly, the *Piezo* + *US* group demonstrated better collagen fibril structure similar to native tissue after 2 months.

Figure 5a | Piezoelectric hydrogel induces cartilage healing, evaluated by histology assessment and mechanical testing *in vivo*. a, Collagen II staining to evaluate the articular cartilage regeneration for sham (defect only), non-piezo/piezo hydrogels with and without US activation (1-2 months), scale bar: 500 μm .

To further investigate the hypertrophic chondrocyte cells, we also conducted collagen X staining. **Supplementary Figure 8** (or Figure below) illustrates that in the *Piezo* + *US* group, both at 1- and 2-month time points, the newly formed tissues mostly appeared in the background color (lavender), indicating the absence of collagen X (dark brown). Noticeably, the *Piezo* group without US showed non-collagen X tissue at the 1-month time point, but after 2 months, the tissue transformed into hypertrophic cartilage. Meanwhile, the other groups showed highly positive collagen X staining in the newly formed tissue both at 1 or 2 month time points.

Regarding collagen I, we consider it as a marker for evaluating osteogenic features to distinguish between bone and cartilage. However, we can also differentiate bone and cartilage using Safranin-O staining, where red color indicates cartilage and blue indicates bone. Therefore, this information is already provided in **Figure 5a**. Hence, we believe conducting collagen I staining experiment in this study is optional.

Supplementary Figure 8 | Piezoelectric hydrogel for cartilage healing evaluated by chondrocyte hypertrophy. Collagen X staining to evaluate the hypertrophic chondrocyte for sham (defect only), non-piezo/piezo hydrogels with and without US activation (1-2 months). Collagen X was identified as dark brown color and Non-collagen X, (background) was identified as lavender color. The

Piezo + US group, both at 1- and 2-month time points, the newly formed tissues mostly appeared in the background color. Meanwhile, the other group showed highly positive collagen X staining in the newly formed tissue both at 1 or 2 month time points, scale bar: 500 μm .

c. We believe US stimulation is applicable on human for following reasons:

First, as observed in the study, the application of 40 kHz US at an intensity of 0.33 Watt/cm² on rabbit knees for a period of 1 or 2 months did not result in any harmful side effects. The rabbits exhibited normal behavior, including walking, and eating (**Supplementary Movie 4**). Therefore, we believe that US conditions employed in the treatment are safe for the treated animals.

Second, it is worth noting that ultrasound therapy has already been widely used for various medical applications in humans [6] including joint pain management [7,8] and bone healing (e.g.:AccelStim)[9]. As responded to your comment #3, the US with a low intensity (< 500 mW/cm²) has been shown to be safe for many clinical uses on human patients.

These together demonstrate that ultrasound has proven its safety and can be used in human applications.

Our current study was only tested on rabbits as proof of concept. However, rabbit cartilage layer is much thinner (250 – 700 μm) and less body weight, compared to humans (e.g.: cartilage layer of 1 - 3 mm). Therefore, a study with a large animal model such as sheep or horses which have a similar cartilage thickness and body weight to the human should be performed. In addition, the condition of US (intensity, frequency, and treatment time) applying on larger animals or humans may need to be adjusted because of the anatomy and physiology differences.

We have revised our writing which included relevant information and these discussions to improve the manuscript.

1. Rossi, A. G. et al. Cyclin-dependent kinase inhibitors enhance the resolution of inflammation by promoting inflammatory cell apoptosis. *Nature Medicine* 12, 1056-1064 (2006). <https://doi.org/10.1038/nm1468>
2. Chu, C. R., Izzo, N. J., Papas, N. E. & Fu, F. H. In Vitro Exposure to 0.5% Bupivacaine Is Cytotoxic to Bovine Articular Chondrocytes. *Arthroscopy: The Journal of Arthroscopic & Related Surgery* 22, 693-699 (2006). <https://doi.org/https://doi.org/10.1016/j.arthro.2006.05.006>
3. Bejarano, F., Lucas, M., Wallace, R., Spadaccino, A. M., & Simpson, H. (2015). Ultrasonic Cutting Device for Bone Surgery Based on a Cymbal Transducer. *Physics Procedia*, 63, 120-126. <https://doi.org/https://doi.org/10.1016/j.phpro.2015.03.020>
4. Chu, C. R., Izzo, N. J., Papas, N. E., & Fu, F. H. (2006). In Vitro Exposure to 0.5% Bupivacaine Is Cytotoxic to Bovine Articular Chondrocytes. *Arthroscopy: The Journal of Arthroscopic & Related Surgery*, 22(7), 693-699. <https://doi.org/https://doi.org/10.1016/j.arthro.2006.05.006>
5. Elmore, S. (2007). Apoptosis: A Review of Programmed Cell Death. *Toxicologic Pathology*, 35(4), 495-516. <https://doi.org/10.1080/01926230701320337>
6. de Lucas, B., Pérez, L. M., Bernal, A. & Gálvez, B. G. Ultrasound Therapy: Experiences and Perspectives for Regenerative Medicine. *Genes* 11, 1086 (2020).
7. Aiyer, R. et al. Therapeutic Ultrasound for Chronic Pain Management in Joints: A Systematic Review. *Pain Medicine* 21, 1437-1448 (2019). <https://doi.org/10.1093/pm/pnz102>

8. di Biase, L. *et al.* Focused Ultrasound (FUS) for Chronic Pain Management: Approved and Potential Applications. *Neurol Res Int* **2021**, 8438498 (2021).
<https://doi.org/10.1155/2021/8438498>
9. Manual, I. AccelStim™. *MEDICAL-ULTRASOUND* **5**, E522288

Once again, we sincerely thank the reviewers for providing helpful comments and suggestions which allowed us to improve the manuscript. If you have any other questions and comments, please let me know.

Sincerely,

Thanh D. Nguyen

Reviewers' Comments:

Reviewer #1:

Remarks to the Author:

The authors have carefully addressed all questions proposed by reviewers, and I believe it is acceptable for publication now.

Reviewer #2:

Remarks to the Author:

In this revised manuscript, the authors have cleared all of the concerns raised by the reviewers, I am having only one question based on the response file, in which it was mentioned that in order to measure the piezoelectric signal from the composite hydrogels, they have to be dried before measurement. So if the wet hydrogel is too conductive to obtain piezoelectric signals, how can they be actually used since the in vivo environment is also wet? Will this wet condition compromise the effectiveness of the piezoelectric effect?

Reviewer #3:

Remarks to the Author:

The Authors have satisfactorily addressed all my concerns.

Thanh D. Nguyen, Associate Professor
University of Connecticut
Department of Mechanical Engineering
Room 356 UTEB, Unit 3139 Storrs, CT 06269
nguyentd@uconn.edu · (860) 486-2415

August 11, 2023

Re: Revision for Manuscript, Nature Communications NCOMMS-23-11040B

Dear Reviewers:

Thank you for your time and consideration on our manuscript. We have updated the material and method in manuscript to reflect the feedback (marked in the red color) and also provided explanations to the questions. Please refer to our point-to-point responses below and the revised manuscript.

Reviewer #1:

Comments: The authors have carefully addressed all questions proposed by reviewers, and I believe it is acceptable for publication now.

Response: Thank you very much.

Reviewer #2:

Comments: In this revised manuscript, the authors have cleared all of the concerns raised by the reviewers, I am having only one question based on the response file, in which it was mentioned that in order to measure the piezoelectric signal from the composite hydrogels, they have to be dried before measurement.

- a) So if the wet hydrogel is too conductive to obtain piezoelectric signals, how can they be actually used since the in vivo environment is also wet?
- b) Will this wet condition compromise the effectiveness of the piezoelectric effect?

Response: We thank the reviewer for the feedback. Please see our clarification and justification for each of your concerns below.

- a) We want to make it clear that the conductivity of the wet hydrogel only prevents the measurement of piezoelectric scaffolds but does not interfere with piezoelectric signal (i.e. local charge) generated for cells to sense (**Figure 3.a**). This is due to the fact that two electrodes of a sensor composed from the wet hydrogel would be short-circuited and the measurement instrument is unable to detect any signal from the short-circuited sensor. Therefore, to assess the piezoelectric performance of the hydrogel, we need to use dried hydrogel. However, the conductivity of the hydrogel does not affect local piezoelectric charges. This is because the nanofibers always generate the electrical charge under ultrasound stimulation (due to the internal atomic displacement to produce internal electrical dipole), regardless of whether the hydrogel is in dried or wet form.

As stem cells are in direct contact with the piezoelectric fibers, they can sense and respond to such local charge at localized regions. We have described and discussed the mechanism of this local charge on inducing chondrogenesis as seen in **Figure 3**.

- b) The wet condition will not affect the piezoelectric effect of the piezoelectric nanofibers in our hydrogel. The atomic displacements inside the PLLA nanofibers under the applied acoustic pressure (ultrasound) leads to the generation of internal dipole inside the nanofibers, generating local charges around the nanofibers. Thus, the piezoelectric materials always generate local charge (i.e. being self-charged) whenever there is a mechanical activation. Also, we have shown the PLLA piezoelectric materials have been commonly used in wet *in vivo* conditions for different purposes including bone, cartilage, skin regeneration or bacterial killing [1-3]. Furthermore, the stem cells are directly integrated within the piezoelectric hydrogel matrix therefore the stem cells can sense any local charges generated by the piezoelectric nanofibers inside the hydrogel. This phenomenon is certified by our *in vitro* and *in vivo* data (**Figure 2, 4 and 5**).

We have incorporated these revisions into the manuscript where information is relevant.

- 1 Liu, Y. et al. Exercise-induced piezoelectric stimulation for cartilage regeneration in rabbits. *Science translational medicine* 14, eabi7282 (2022).
- 2 Liu, D., et al., Ultrasound-triggered piezocatalytic composite hydrogels for promoting bacterial-infected wound healing. *Bioactive Materials*, 2023. 24: p. 96-111
- 3 Das, R., et al., Biodegradable nanofiber bone-tissue scaffold as remotely-controlled and self-powering electrical stimulator. *Nano Energy*, 2020. 76: p. 105028.

Reviewer #3:

Comments: The Authors have satisfactorily addressed all my concerns.

Response: Thank you very much.

Once again, we extend our sincere gratitude to the reviewers for their comments, which have enabled us to refine our manuscript. Please do not hesitate to reach out if you have any questions.

Sincerely,

Thanh D. Nguyen

Reviewers' Comments:

Reviewer #2:

Remarks to the Author:

This revised manuscript can be accepted for publication in its current form.